# Opposing action of the FLR-2 glycoprotein hormone and DRL-1/FLR-4 MAP kinases balance p38-mediated growth and lipid homeostasis in *C. elegans*

Sarah K. Torzone[1,2], Aaron Y. Park[1], Peter C. Breen[1], Natalie R. Cohen[1], Robert H. Dowen [1,2,3]*

**1** Integrative Program for Biological and Genome Sciences, The University of North Carolina at Chapel Hill, Chapel Hill, North Carolina, United States of America, **2** Department of Biology, The University of North Carolina at Chapel Hill, Chapel Hill, North Carolina, United States of America, **3** Department of Cell Biology and Physiology, The University of North Carolina at Chapel Hill, Chapel Hill, North Carolina, United States of America

* dowen@email.unc.edu

**Data Availability Statement:** All relevant data are within the paper and its Supporting Information files.

## Abstract

Animals integrate developmental and nutritional signals before committing crucial resources to growth and reproduction; however, the pathways that perceive and respond to these inputs remain poorly understood. Here, we demonstrate that DRL-1 and FLR-4, which share similarity with mammalian mitogen-activated protein kinases, maintain lipid homeostasis in the *C. elegans* intestine. DRL-1 and FLR-4 function in a protein complex at the plasma membrane to promote development, as mutations in *drl-1* or *flr-4* confer slow growth, small body size, and impaired lipid homeostasis. To identify factors that oppose DRL-1/FLR-4, we performed a forward genetic screen for suppressors of the *drl-1* mutant phenotypes and identified mutations in *flr-2* and *fshr-1*, which encode the orthologues of follicle stimulating hormone and its putative G protein–coupled receptor, respectively. In the absence of DRL-1/FLR-4, neuronal FLR-2 acts through intestinal FSHR-1 and protein kinase A signaling to restrict growth. Furthermore, we show that opposing signaling through DRL-1 and FLR-2 coordinates TIR-1 oligomerization, which modulates downstream p38/PMK-1 activity, lipid homeostasis, and development. Finally, we identify a surprising noncanonical role for the developmental transcription factor PHA-4/FOXA in the intestine where it restricts growth in response to impaired DRL-1 signaling. Our work uncovers a complex multi-tissue signaling network that converges on p38 signaling to maintain homeostasis during development.

## Introduction

Animals respond to environmental, nutritional, and developmental cues to balance resources between essential biological processes, ensuring fitness and reproductive fidelity. In metazoans, reproduction is a metabolically expensive process, requiring organisms to shift somatic

**Funding:** This work was supported by the National Institute of General Medical Sciences grant R35GM137985 to R.H.D. The funders had no role in study design, data collection and analysis, decision to publish, or preparation of the manuscript.

**Competing interests:** The authors have declared that no competing interests exist.

**Abbreviations:** AID, auxin-inducible degron; FSH, follicle-stimulating hormone; MAPK, mitogen-activated protein kinase; PKA, protein kinase A; TIR, Toll/interleukin-1 receptor; VLDL, very low-density lipoprotein.

energy stores to the germline to support the development of their offspring. This metabolic trade-off ensures reproductive fitness while restricting the somatic maintenance programs that support longevity [1,2]. The energetic balance between somatic and germline functions is coordinated by complex regulatory networks across diverse tissues that integrate developmental and environmental inputs; however, the homeostatic mechanisms that govern these metabolic trade-offs are not fully understood.

In many metazoans, including the nematode *Caenorhabditis elegans*, development into a reproductive adult is marked by production of vitellogenin proteins, which are structural and functional orthologues of the mammalian apoB protein that coordinates very low-density lipoprotein (VLDL) assembly, secretion, and reabsorption in the liver [3]. In *C. elegans*, the vitellogenins package intestinal lipids into VLDL-like particles, which are then secreted and captured by the LDL receptor RME-2 in oocytes [4,5]. The vitellogenin-associated lipids promote the recruitment of sperm to the oocyte during fertilization [6], support robust development of the progeny [7], and facilitate larval survival during starvation conditions [8,9]. While crucial for reproduction and the developmental success of the progeny, reallocation of these key lipid resources restricts maternal longevity [10,11]. Consistently, this metabolic trade-off can be finely tuned and is highly regulated by developmental, nutritional, and metabolic regulatory pathways [10,12–16]. The molecular basis of how these developmental regulators impact metabolic decisions to maintain organismal homeostasis is poorly understood.

Genetic screens aimed at uncovering genes required for the initiation of vitellogenesis have identified proteins with broader roles in development, metabolism, stress responses, and longevity [9,14,16]. Furthermore, impaired vitellogenesis can dramatically alter intestinal lipid levels, and conversely, metabolic dysfunction can down-regulate vitellogenin production, with either event resulting in a defect in overall lipid homeostasis. We previously identified the dietary-restriction-like gene *drl-1* in an RNAi screen as a candidate regulator of vitellogenesis [14]. DRL-1 is a serine–threonine mitogen-activated protein kinase (MAPK) orthologous to mammalian MEKK3 that has been implicated in regulating metabolic, detoxification, and aging pathways [17,18]. Loss of *drl-1* increases life span and up-regulates detoxication genes, which requires the p38 MAPK signaling pathway (NSY-1/SEK-1/PMK-1); however, the dietary restriction-like metabolic state triggered by *drl-1* knockdown is not entirely dependent on p38 signaling [18]. Interestingly, loss of a closely related MAP kinase gene, *flr-4*, induces a similar p38-dependent life span extension and induction of detoxication genes [19,20]. This observation suggests that DRL-1 and FLR-4 may function in the same signaling pathway; however, a biochemical association between these two proteins has not been demonstrated.

While the role of the p38/PMK-1 pathway in regulating innate immunity and oxidative stress responses is well defined [21–23], its function in development, as well as the molecular pathways that converge on p38 signaling to coordinate growth, are poorly understood. Here, we find that mutations in *drl-1* or *flr-4* severely impair development, growth, and lipid homeostasis in *C. elegans*, in part by governing the oligomerization and activation of TIR-1/SARM1, a Toll/interleukin-1 receptor (TIR) domain-containing protein that activates p38/PMK-1 signaling [24]. We show that DRL-1 and FLR-4 are opposed by glycoprotein hormone signaling, which is mediated by the secreted neurohormone FLR-2, its putative intestinal G protein–coupled receptor FSHR-1, and downstream cAMP/protein kinase A (PKA) signaling. Moreover, our data suggest that these opposing pathways may converge on TIR-1 in the intestine to modulate p38 signaling and govern the subcellular localization of the PHA-4/FOXA transcription factor, a well-established regulator of development, to control growth and metabolic homeostasis. Thus, we demonstrate that intestinal p38/PMK-1 activity is coordinated by a non-cell-autonomous hormonal signal and intestinal MAPK pathway to maintain metabolic homeostasis and ensure robust development.

## Results

### The DRL-1 protein kinase functions in the intestine to modulate fat transport and growth

The vitellogenin genes (*vit-1* through *vit-6*) are specifically expressed in the *C. elegans* intestine during the L4 larval to adult transition, coinciding with the onset of reproduction. This metabolic commitment can be precisely followed using a multi-copy P*vit-3*::*GFP* or a single-copy P*vit-3*::*mCherry* reporter transgene comprised of GFP or mCherry under the control of the *vit-3* promoter, respectively (Figs 1A and S1A). Notably, the single-copy P*vit-3*::*mCherry* transgene is highly sensitive and its expression closely resembles endogenous *vit-3* expression, and thus, it is the primary vitellogenesis reporter used in this study.

We previously performed an RNAi screen for factors that are required for proper vitellogenin expression at the onset of adulthood and identified the *drl-1* gene as essential for robust P*vit-3*::*GFP* expression [14]. The *drl-1* gene encodes a protein kinase with highest similarity to the mammalian mitogen-activated protein kinase kinase kinase 3 (MAP3K3/MEKK3) protein [17]. To eliminate the possibility that *drl-1* RNAi produces off-target effects that impairs vitellogenesis, we generated several *drl-1* genetic mutants, which are all likely full loss-of-function alleles, using CRISPR/Cas9 gene editing and assessed P*vit-3* reporter expression. Indeed, loss of *drl-1* resulted in a dramatic reduction in vitellogenin reporter expression (Figs 1A, S1A and S1B). These findings are consistent with the observation that knockdown of *drl-1* reduces intestinal lipid stores [17], which could impair lipoprotein synthesis and assembly. Vitellogenin gene expression is regulated through non-cell-autonomous and cell-autonomous mechanisms via hypodermal, germline, and intestinal regulators [25], which prompted us to test where *drl-1* functions in the worm to regulate *vit* gene expression. After introduction of an auxin-inducible degron (AID) tag into the endogenous *drl-1* locus using CRISPR/Cas9 genome editing [26,27], we performed tissue-specific DRL-1 protein depletion in hypodermal or intestinal cells and assessed P*vit-3*::*mCherry* reporter expression. DRL-1 depletion in the intestine, but not the hypodermis, markedly impaired reporter expression (Figs 1B and S2), indicating that DRL-1 functions cell-autonomously to control vitellogenesis. Moreover, knockdown of *drl-1* in the intestine, but not in the hypodermis, using tissue-specific RNAi reduced the expression of the endogenous *vit* genes (Fig 1C). Consistently, we were able to rescue vitellogenin reporter expression in *drl-1* mutant animals with a transgene that expresses *drl-1* under the control of an intestinal, but not a hypodermal, promoter (S1C Fig), indicating that intestinal *drl-1* expression is sufficient to restore vitellogenin expression.

Impaired lipid homeostasis can have profound effects on organismal growth and many mutants with vitellogenesis phenotypes also display gross defects in development [25]. Thus, we inspected the *drl-1* mutants for growth rate and body size phenotypes, finding severe defects (S1D, S1E and S2C Figs). Consistent with our previous observations, genetic rescue of the *drl-1* mutant with intestinal, but not hypodermal, *drl-1* expression strongly suppressed these developmental phenotypes (Fig 1D and 1E). Together, these data demonstrate an intestinal role for DRL-1 in the regulation of lipid allocation, growth rate, and body size.

### Intestinal DRL-1 and FLR-4 interact to form a presumptive protein kinase complex

The *flr-4* gene encodes a MAP kinase that is required for metabolic homeostasis and proper aging, and importantly, mutants display a diet-specific life span extension that is similar to that of *drl-1* mutants [17,20], suggesting that these two MAP kinases may act together to govern lipid homeostasis, growth, and development. To investigate this possibility, we first

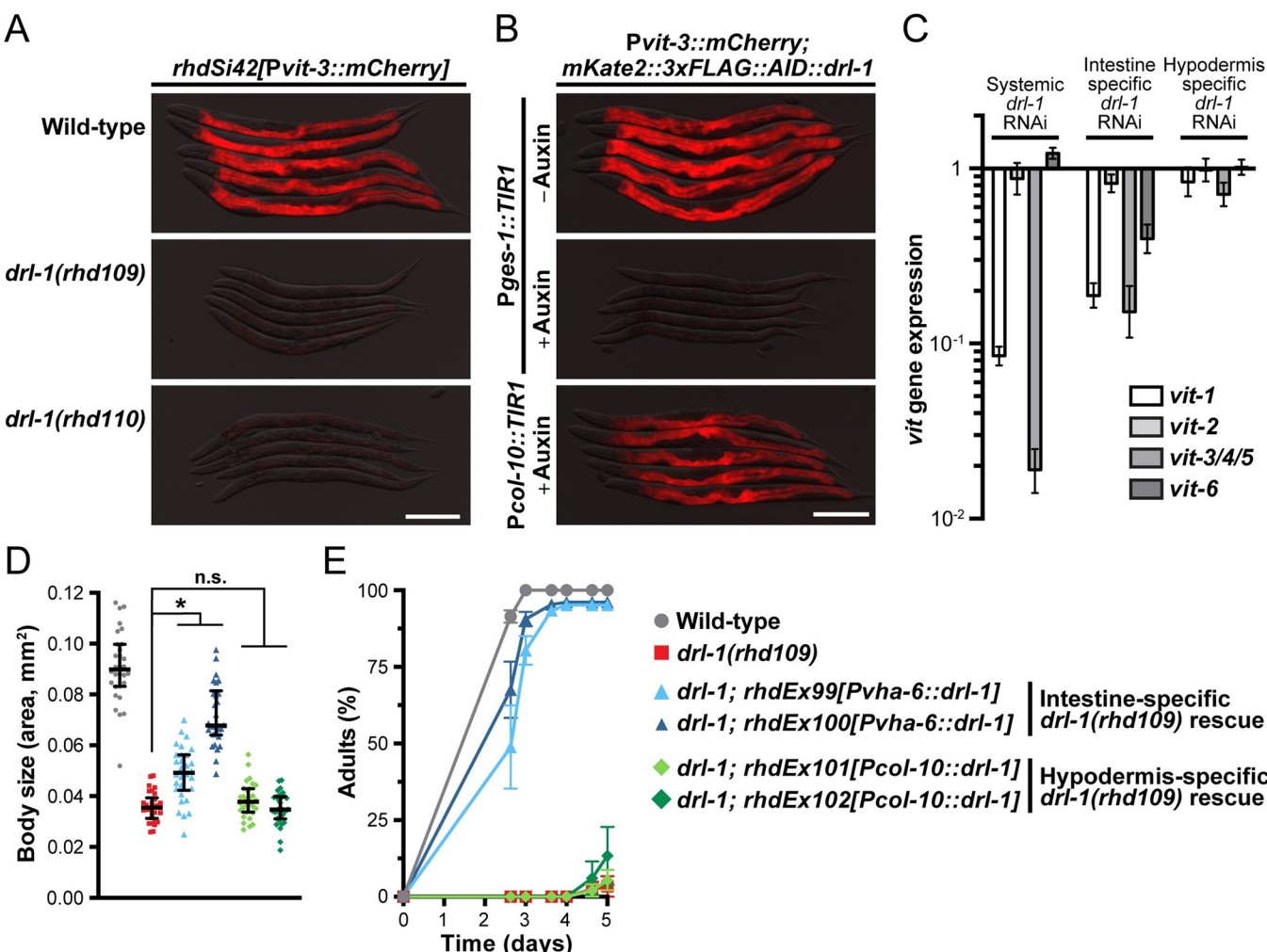

**Fig 1. DRL-1 functions in the intestine to promote development, growth, and lipid reallocation.** Representative overlaid DIC and mCherry fluorescence images of day 1 adult animals expressing a single-copy P*vit-3*::*mCherry* reporter (*vit-3* promoter fused to *mCherry*) in (**A**) wild-type or *drl-1* mutant animals or (**B**) *mKate2*::*3xFLAG*::*AID*::*drl-1* animals following intestinal (P*ges-1*::*TIR1*) or hypodermal (P*col-10*::*TIR1*) degradation using 4 mM auxin (scale bars, 200 μm). (**C**) RT-qPCR analysis of endogenous *vit* gene expression in day 1 adult animals after whole-body or tissue-specific knockdown of *drl-1* by RNAi. (**D**) Body size and (**E**) growth rate of wild-type, *drl-1(rhd109)*, or the indicated *drl-1(rhd109)* tissue-specific rescue strains (2 independent lines each). All strains contain *mgIs70* and the data are presented as (D) the median and interquartile range (*, $P < 0.0001$, one-way ANOVA) or (E) the mean +/− SEM of 3 independent experiments. Raw data underlying panels C, D, and E can be found in S1 Data.

inspected the kinase domain of DRL-1 and FLR-4, as well as several other similar MAPKs, and discovered that while DRL-1 shares significant sequence similarity in the kinase domain, it lacks several of the conserved amino acids that participate in ATP binding and catalysis (Fig 2A). To address whether DRL-1 possesses kinase activity, we used CRISPR/Cas9 to generate a DRL-1 P269S mutation, which is analogous to the FLR-4 P223S hypomorphic mutation that is positioned within the activation loop of the kinase domain and yields temperature-sensitive phenotypes [19]. Similar to the *flr-4(P223S)* mutant, *drl-1(P269S)* animals display temperature-sensitive defects in vitellogenin expression (Figs 2B and S3A). Additional mutation of E253 and G254 in the kinase domain, which are required for in vitro kinase activity [17], enhanced the temperature-sensitive vitellogenesis phenotype to similar levels as the null mutants (S3A Fig). Not only is the presumptive kinase activity of DRL-1 and FLR-4 necessary for proper *vit* gene expression, but it is required to suppress the expression of the β-oxidation

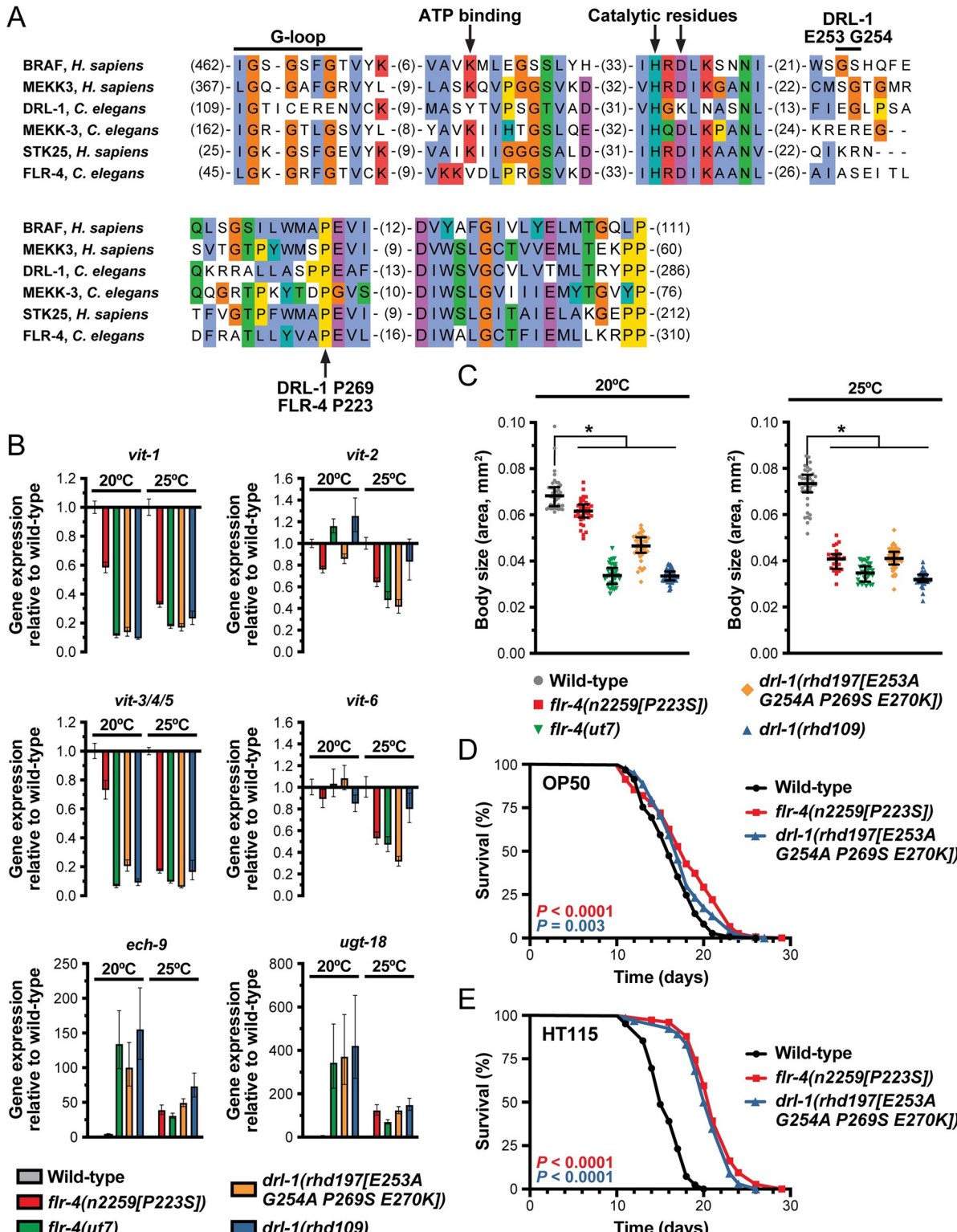

**Fig 2. The presumptive kinase activity of DRL-1 and FLR-4 is essential for function.** (**A**) An amino acid alignment (MUSCLE) of the kinase domain of the indicated MAP kinases. The key residues are indicated, including those mutated in the presumptive kinase dead mutants (DRL-1 E253, G254, P269; FLR-4 P223). (**B**) Expression of the indicated genes as measured by RT-qPCR in day 1 adult animals grown at 20 or 25°C. (**C**) Body size of the indicated strains at 20 or 25°C (*, *P* < 0.0001, one-way ANOVA). (**B, C**) The *drl-1(rhd109)* and *flr-4(ut7)* alleles are presumed to be null mutations. (**D, E**) Longitudinal life span assays of wild-type and presumptive kinase dead mutants

reared at 25°C with FUDR on *E. coli* (**D**) OP50 or (**E**) HT115. Log-rank test *P* values are reported. Raw data underlying panels B, C, D, and E can be found in S2 Data.

gene *ech-9* and the detoxification gene *ugt-18* (Fig 2B) [17], indicating that DRL-1 and FLR-4 function broadly in regulation of intestinal metabolism.

Similar to the *drl-1* null mutants, *flr-4* null mutants display severely reduced developmental rates and body size defects when reared on *E. coli* OP50, the standard laboratory diet (S3B and S3C Fig) [20,28]. Furthermore, the presumptive kinase activity of DRL-1 and FLR-4 is required to maintain proper body size (Fig 2C), suggesting that these kinases may function in a broader signaling cascade to balance metabolic needs during development and aging. Intriguingly, the FLR-4 P223S activation loop mutant exhibits a life span extension only when reared on *E. coli* HT115, and not OP50, suggesting that components in the diet interface with longevity pathways downstream of FLR-4 [20]. To investigate whether DRL-1 also restricts life span in a diet-dependent manner, we performed longitudinal life span assays on the *flr-4(P223S)* and *drl-1 (P269S)* mutants fed different *E. coli* food sources. Indeed, the presumptive DRL-1 and FLR-4 kinase dead mutants are markedly long-lived on *E. coli* HT115 (Fig 2D and 2E). These findings argue that the kinase activity of both FLR-4 and DRL-1 are required to restrict longevity in response to diet. Consistently, overexpression of *drl-1* on an extrachromosomal array is sufficient to suppress the growth rate defects induced by simultaneous loss of both *drl-1* and *flr-4* (S3D Fig), suggesting that DRL-1, and its potential kinase activity, can compensate for loss of *flr-4*. Together, these data suggest that the kinase activity of DRL-1 and FLR-4 is required to maintain overall organismal homeostasis; however, we cannot rule out the possibility one or both proteins lack kinase activity and act as a non-catalytic partner to another protein kinase.

These observations support the intriguing possibility that DRL-1 and FLR-4 act in the same tissue (i.e., the intestine), or even potentially in a protein complex, to coordinate developmental or nutritional programs. To investigate this hypothesis, we first sought to define where FLR-4 functions, focusing on the intestinal and neuronal tissues [19,20]. Using CRISPR/Cas9 to introduce an AID tag at the endogenous *flr-4* locus, we performed tissue-specific depletion experiments and assessed vitellogenin expression, body size, and growth rate. Depletion of FLR-4 in the intestine, but not in neurons, abrogated P*vit-3*::*mCherry* expression, severely reduced body size, and dramatically slowed growth rates (Figs 3A–3C and S4A). Surprisingly, post-developmental depletion of either DRL-1 or FLR-4 reduced P*vit-3*::*mCherry* reporter expression and decreased the body size of day 2 adults (S4B–S4F Fig), indicating that these phenotypes are not solely a consequence of impaired development and are likely driven by metabolic dysfunction. Finally, intestinal depletion of either DRL-1 or FLR-4 is sufficient to confer unique responses to different *E. coli* food sources, including HT115-dependent effects on growth rate and life span (S5 Fig). Taken together, these data indicate that FLR-4 acts cell-autonomously, during both larval stages and adulthood, to regulate intestinal homeostasis, possibly by directly interacting with DRL-1.

To test whether DRL-1 and FLR-4 colocalize, we used CRISPR/Cas9 gene editing to generate a strain that coexpresses HA::mGreenLantern::FLR-4 and mKate2::3xFLAG::DRL-1. Although these proteins are expressed at low levels, we detected the mGL::FLR-4 and mKate2::DRL-1 proteins at the intestinal plasma membrane (Fig 3D), consistent with overexpression studies of FLR-4 and DRL-1 [29,30]. Furthermore, reciprocal co-immunoprecipitation studies demonstrated that HA::FLR-4 and FLAG::DRL-1 interact (Fig 3E), likely forming a larger protein kinase complex. These data explain why *drl-1* and *flr-4* mutants have similar phenotypes and provide a mechanistic basis for their role in maintaining intestinal homeostasis via MAPK signaling.

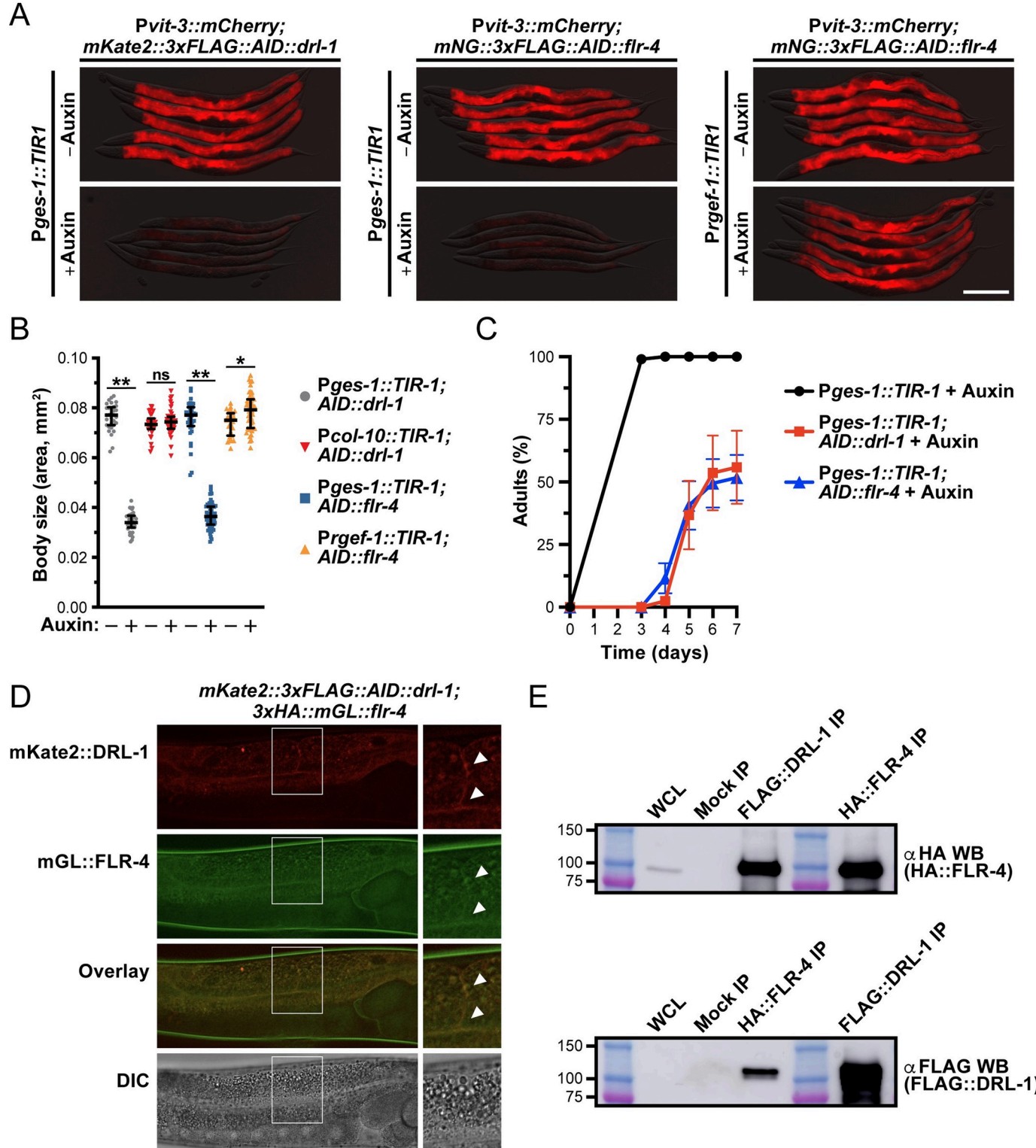

**Fig 3. Intestinal DRL-1 and FLR-4 function in a complex to promote growth and lipoprotein production.** (**A-C**) Phenotypic characterization of *mKate2::3xFLAG::AID::drl-1* or *mNG::3xFLAG::AID::flr-4* animals carrying the P*vit-3::mCherry* reporter after tissue-specific depletion with 4 mM auxin (P*ges-1::TIR1*, intestinal depletion; P*rgef-1::TIR1*, pan-neuronal depletion; P*col-10::TIR1*, hypodermal depletion). (**A**) Representative overlaid DIC and mCherry fluorescence images of day 1 adults with or without auxin treatment (scale bar, 200 μm), (**B**) body size of day 1 adults (median and interquartile range; ns, not significant, *, $P = 0.003$, **, $P < 1 \times 10^{-45}$, $t$ test), and (**C**) growth rate (mean +/− SEM) of animals after tissue-specific protein depletion. (**D**) Colocalization of

mKate2::3xFLAG::DRL-1 and 3xHA::mGL::FLR-4 in intestinal cells of *glo-4(ok623)* animals (scale bar, 50 μm). Panels on the right show magnified images of the outlined regions and arrowheads point to areas of strong colocalization. (**E**) Co-immunoprecipitation of the indicated proteins after mock, anti-FLAG, or anti-HA immunoprecipitation followed by western blotting. As a positive control, immunoprecipitations were probed with the same antibodies (far right lanes). The whole cell lysate (WCL) represents 5% of the IP input. FLAG::DRL-1 was not detectable in the WCL likely due to low levels of protein expression or poor sensitivity of the anti-FLAG antibody. The co-IP/western blot experiment was performed twice with similar results. Raw data underlying panels B and C can be found in S3 Data, and raw images for panel E can be found in S1 Raw Images.

## The FLR-2 neuropeptide hormone antagonizes DRL-1 and FLR-4 signaling

DRL-1/FLR-4 signaling promotes organismal development, growth, and aging. Yet, it is possible that pro-growth signaling through this pathway may be dynamically tuned in response to adverse environmental or nutritional conditions to temper development. It is likely that this signaling would need to be balanced by other signaling events to maintain overall homeostasis. Thus, we reasoned that in the absence of *drl-1/flr-4*, other pathways may actively restrict development. To identify components of this signaling axis, we performed a forward genetic screen to isolate mutations that suppress the growth and vitellogenesis defects associated with the *drl-1(rhd109)* mutant. We isolated over 100 mutants and initially pursued a small pilot set for further analysis. Following backcrossing and whole genome sequencing, we identified a putative null mutation in the *flr-2* gene as a likely *drl-1* suppressor mutation (S1 Table).

The *flr-2* gene encodes a neuronally expressed secreted protein with highest similarity to the human glycoprotein hormone subunit α2 (GPA2) of thyrostimulin [31,32], as well as weaker similarity to the α subunits of other human glycoprotein hormones (i.e., FSH, LH, and TSH). The *flr-2(rhd117)* mutation identified in our screen, as well as a second allele *flr-2(ut5)*, strongly suppressed the defects in vitellogenin reporter expression observed in the *drl-1* and *flr-4* mutants (Figs 4A and S6A). Furthermore, while loss of *drl-1* impaired the accumulation of neutral lipids as revealed by reduced Oil Red O staining [17], the *drl-1; flr-2* double mutants displayed wild-type fat levels (Figs 4B and S6B). Consistent with these defects in vitellogenesis and lipid homeostasis, the *drl-1* mutant animals also had a dramatically smaller brood size; however, *drl-1; flr-2* double mutants produced nearly as many progeny as wild-type animals (Fig 4C). Together, these data suggest that FLR-2 opposes MAPK signaling to balance intestinal resources to maintain homeostasis.

To assess whether FLR-2 impairs growth upon loss of *drl-1*, we performed growth measurements in the *drl-1* and *drl-1; flr-2* mutants. Indeed, both *flr-2* alleles suppress the slow growth rate and small body size phenotypes displayed by the *drl-1* mutant (Fig 4D and 4E). Mutations in the nuclear hormone receptor *nhr-49* (orthologous to human HNF4 and PPARα), which has been previously shown to suppress *drl-1* RNAi phenotypes [17], failed to suppress the vitellogenesis, growth rate, and body size defects of the *drl-1(rhd109)* mutant (Fig 4A, 4D and 4E). It is possible that experimental differences, such as the type of food or degree of *drl-1* inactivation (RNAi versus mutant), account for these discrepancies.

Expression of the FLR-2 protein is limited to a small set of neurons in the head and tail [32], suggesting that FLR-2 functions non-cell-autonomously to regulate intestinal homeostasis. Indeed, genetic rescue of *flr-2* under the control of a pan-neuronal promoter partially reverses the growth and body size phenotypes of the *drl-1; flr-2* double mutants (S6C and S6D Fig), suggesting that expression of *flr-2* in neurons is sufficient to slow the growth of *drl-1* mutant animals. To explore whether *flr-2* is developmentally regulated, we inserted a HA tag into the *flr-2* locus using CRISPR/Cas9 and performed a western blot analysis of HA::FLR-2 from lysates of animals at different developmental stages. While FLR-2 is expressed throughout larval development and into adulthood, expression peaks at the L3 stage and is lowest during reproduction (S7A Fig), suggesting that *flr-2* expression may be repressed at adulthood to facilitate vitellogenin production. Notably, knockdown of *drl-1* or *flr-4* did not alter FLR-2

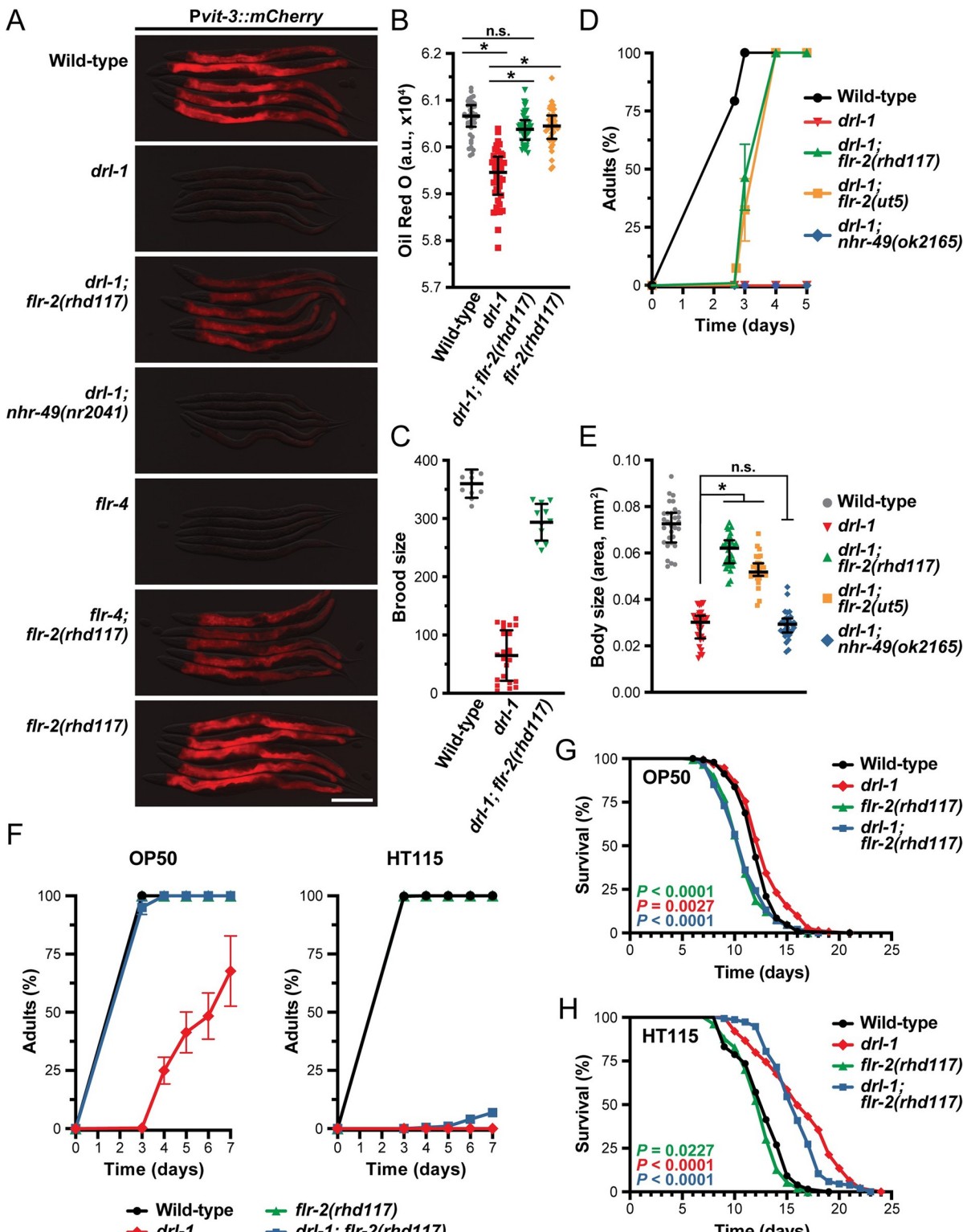

**Fig 4. Loss of *flr-2* suppresses mutations in *drl-1* and *flr-4* when reared on specific food sources.** (**A**) Representative fluorescence images of P*vit-3::mCherry* reporter expression (day 1 adults; scale bar, 200 μm), (**B**) whole animal Oil Red O staining (day 1 adults; median and interquartile range; ns, not significant, \*, $P < 0.0001$, one-way ANOVA), (**C**) brood size (mean +/− SD), (**D**) growth rate (mean +/− SEM), and (**E**) body size (day 1 adults; median and interquartile range; ns, not significant, \*, $P < 0.0001$, one-way ANOVA) for wild-type animals and the indicated mutants grown on *E. coli* OP50. (**D**, **E**) Strains contain the *mgIs70* transgene except for the *drl-1; nhr-49(ok2165)* strain. (**F**)

The *flr-2(rhd117)* mutation robustly suppresses *drl-1* mutant growth defects when animals are reared on *E. coli* OP50 (left), but not *E. coli* HT115 (right). Longitudinal life span assays of wild-type animals and the indicated mutants reared at 25°C with FUDR on *E. coli* (**G**) OP50 or (**H**) HT115. Animals were L4s at day 0 and *P* values (log-rank test) are reported. (**A-H**) The *drl-1(rhd109)* and *flr-4(ut7)* alleles were used in these studies. Raw data underlying panels B, C, D, E, F, G, and H can be found in S4 Data.

levels, and constitutive overexpression of *flr-2* in neurons did not dramatically suppress or enhance the body size defects exhibited by the *drl-1* mutant, which together, indicate that *flr-2* acts in parallel, and not downstream, of *drl-1* (S7B and S7C Fig). Our data demonstrate that the FLR-2 neurohormone is a potent inhibitor of growth, development, and reproduction in the absence of active DRL-1/FLR-4 signaling.

The life span and stress responses of *drl-1* and *flr-4* mutants are markedly different when animals are fed *E. coli* HT115 bacteria compared to OP50, likely due to differences in the nutritional value of the strains [33]. Consistent with these observations, genetic mutation of *drl-1*, as well as intestinal depletion of AID::DRL-1, more severely impairs growth of animals reared on *E. coli* HT115 compared to OP50 (Figs 4F and S5A). Although mutation of *flr-2* suppresses the growth defects of the *drl-1(rhd109)* mutant on OP50, it surprisingly fails to suppress *drl-1 (rhd109)* growth on HT115 (Fig 4F). Similarly, the *flr-2* mutation fails to suppress the longevity conferred by loss of *drl-1* when animals are grown on *E. coli* HT115 (Fig 4G and 4H). Notably, loss of *flr-2*, which results in short-lived animals [32], suppresses the modest life span increase displayed by *drl-1* mutant animals on OP50 (Fig 4G). Our results demonstrate that while loss of *flr-2* strongly suppresses the *drl-1* mutant phenotypes on OP50, it fails to yield similar results on HT115, suggesting that a HT115-specific nutritional input may be acting redundantly with FLR-2 to oppose DRL-1/FLR-4 signaling.

## FLR-2 acts via the G protein–coupled receptor FSHR-1 to stimulate PKA activity

The FLR-2 protein is a secreted hormone that likely acts by binding a cell surface receptor on a distal tissue, possibly the intestine, to stimulate a signaling cascade that slows animal development. We predicted that mutations in the FLR-2 receptor may also suppress the growth and vitellogenesis defects displayed by the *drl-1* mutant. Indeed, sequencing of additional *drl-1* suppressor mutations identified a putative null mutation in the *fshr-1* gene (S2 Table), which encodes a G protein–coupled receptor with similarity to the family of glycoprotein hormone receptors that include the TSH, FSH, and LH receptors in humans [34]. The *fshr-1(rhd118)* mutation, as well as the well-characterized *fshr-1(ok778)* allele, both suppressed the vitellogenesis defects conferred by loss of either *drl-1* or *flr-4* (Figs 5A and S8A). Furthermore, the *fshr-1* mutations suppressed the slow growth and reduced the body size of the *drl-1(rhd109)* mutant (Fig 5B and 5C). The partial suppression of the body size can be attributed to the fact that *flr-2* and *fshr-1* single mutants are smaller than wild-type animals (S6D and S8B Figs). We reasoned that if FSHR-1 is the intestinal receptor for FLR-2, then knockdown of *fshr-1* specifically in the intestine would suppress the *drl-1* mutation. Indeed, intestine-specific knockdown of *fshr-1* suppresses the vitellogenesis defects and small body size of *drl-1* mutant animals to similar levels as systemic *fshr-1* knockdown (Fig 5D and 5E). Together, these data suggest that intestinal FSHR-1 mediates the effects of FLR-2 to slow developmental rate of animals when DRL-1/FLR-4 signaling is reduced.

The mammalian FSH receptor, like other glycoprotein hormone receptors, signals through heterotrimeric G proteins, primarily $G\alpha_s$, to stimulate adenylate cyclase activity, cAMP production, and PKA activation [35,36]. In *C. elegans*, gain-of-function mutations in *gsa-1* ($G\alpha_s$) or *acy-1* (adenylate cyclase) suppresses the germline defects observed in the *fshr-1* mutant,

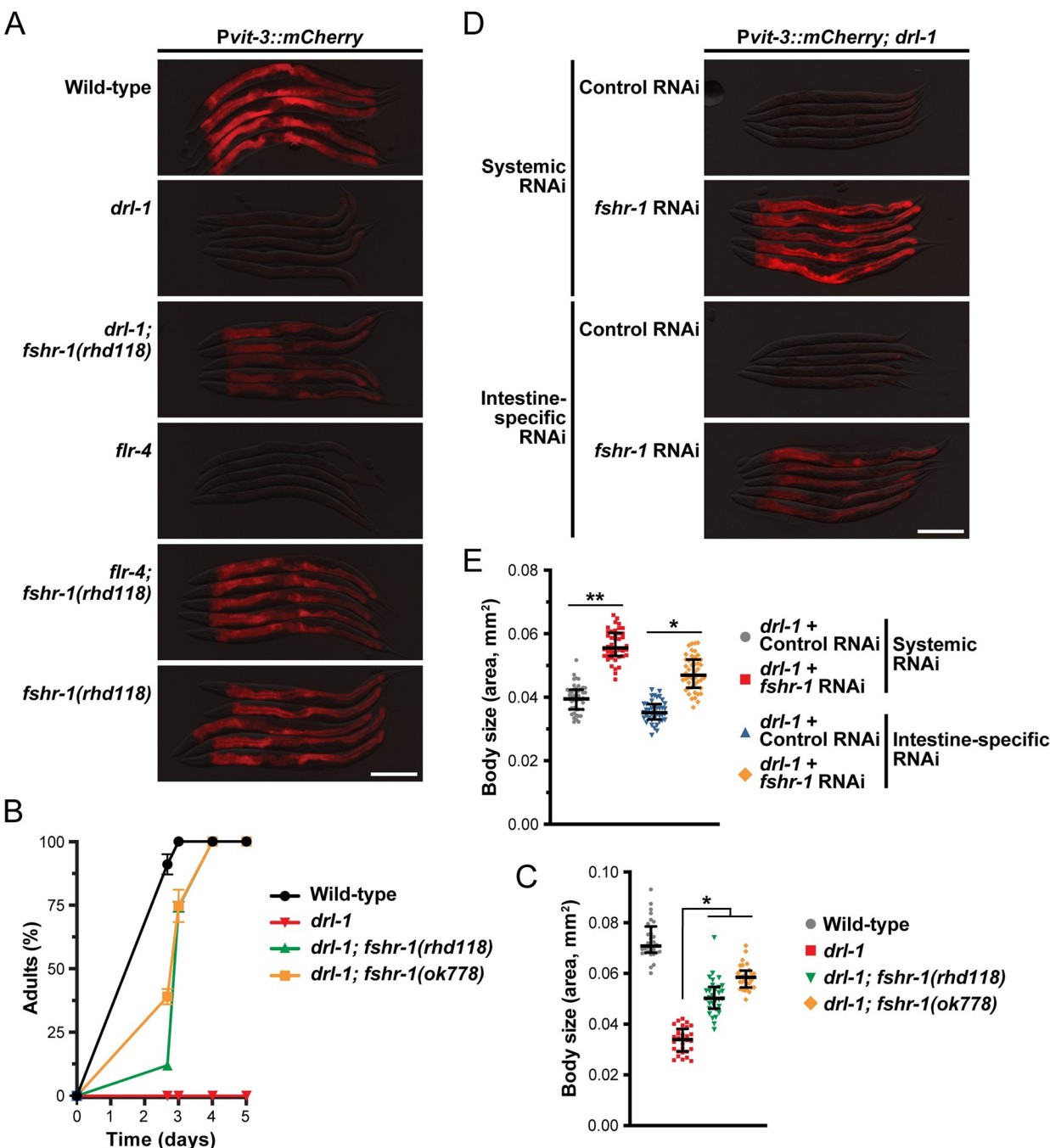

**Fig 5. The intestinal GPCR FSHR-1 opposes DRL-1/FLR-4 signaling.** (**A**) Representative overlaid DIC and mCherry fluorescence images (day 1 adults; scale bar, 200 μm), (**B**) growth rate (mean +/− SEM), and (**C**) body size (day 1 adults; median and interquartile range; *, $P < 0.0001$, one-way ANOVA) for wild-type and mutant animals. (**B, C**) Strains contain the *mgIs70* transgene. (**D**) P*vit-3::mCherry* reporter expression (scale bar, 200 μm) and (**E**) body size (median and interquartile range; *, $P < 1 \times 10^{-19}$, **, $P < 1 \times 10^{-30}$, *t* test) of day 1 adult *drl-1(rhd109)* animals subjected to systemic or intestine-specific RNAi. Raw data underlying panels B, C, and E can be found in S5 Data.

providing genetic evidence that this pathway functions in the worm. Thus, we predicted that loss of *gsa-1*, the *acy-1-4* genes (adenylate cyclase), or *kin-1* (PKA) would also suppress the *drl-1* mutant phenotypes if FSHR-1 couples to this signaling pathway (Fig 6A). RNAi knockdown of *gsa-1*, *acy-4*, or *kin-1* reactivated vitellogenin reporter expression to different degrees in the

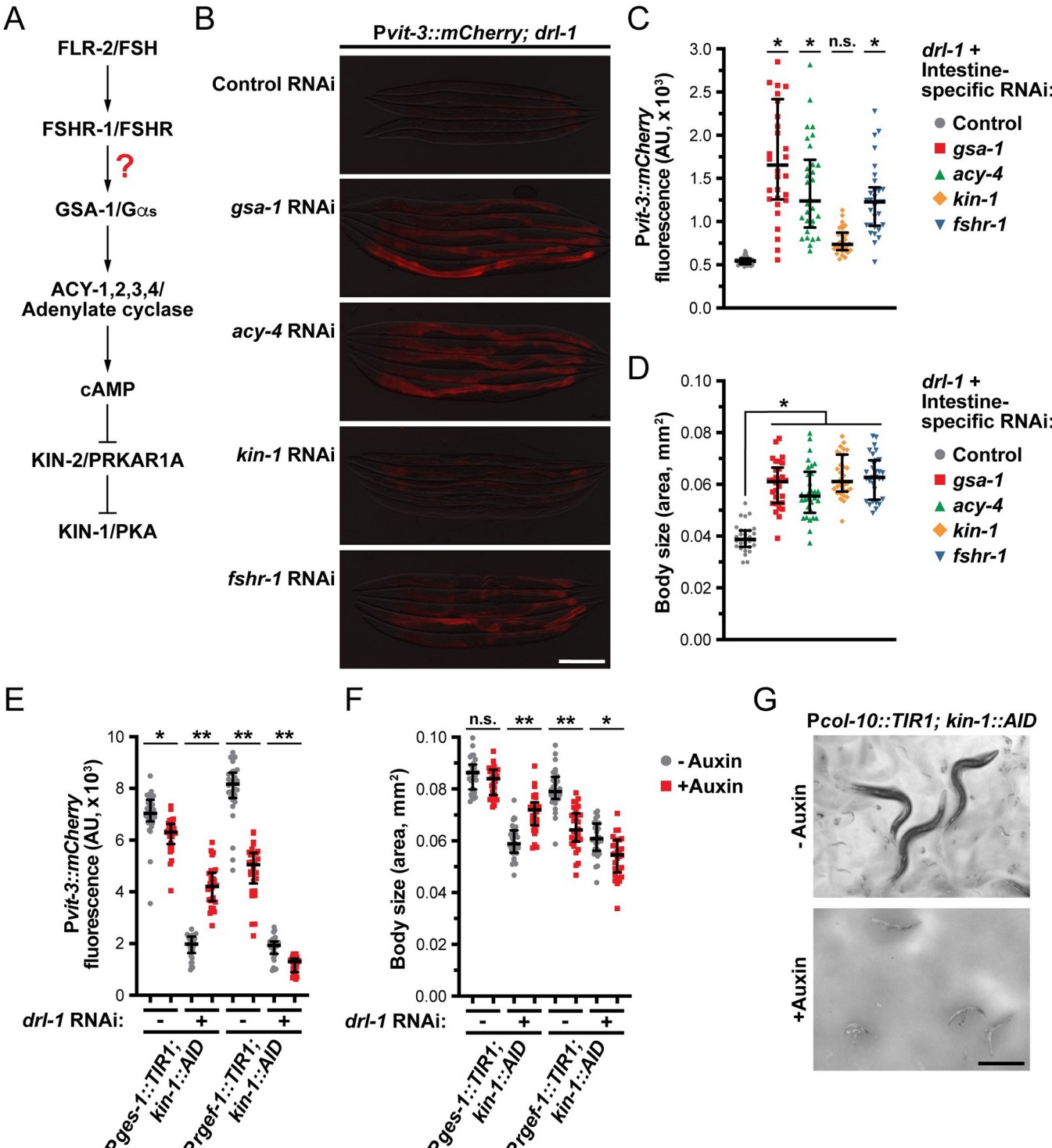

**Fig 6. FLR-2 and FSHR-1 function through intestinal Gα$_s$/cAMP signaling.** (**A**) A diagram of how FLR-2/FSHR-1 may activate PKA via Gα$_s$/cAMP signaling. (**B**) Representative overlaid DIC and mCherry fluorescence images of *drl-1(rhd109)* day 1 adults following systemic RNAi (scale bar, 200 μm). (**C**) Quantification of P*vit-3::mCherry* reporter expression and (**D**) body size of *drl-1(rhd109)* animals subjected to intestine-specific RNAi (day 1 adults; median and interquartile range; ns, not significant, *, $P < 0.0001$, one-way ANOVA). (**C**, **D**) The intestine-specific RNAi was performed for one generation apart from *kin-1*, which was performed for two generations. (**E**) Quantification of P*vit-3::mCherry* reporter expression and (**F**) body size of *kin-1::AID* animals subjected to control or *drl-1* RNAi in the absence or presence of 4 mM auxin (median and interquartile range; n.s., not significant, *, $P < 0.0005$, **, $P < 0.0001$, *t* test). KIN-1 was either depleted in the intestine (P*ges-1::TIR1*) or in neurons (P*rgef-1::TIR1*) using a 24-hour auxin treatment that was initiated at the L4 stage. (**G**)

Hypodermal depletion of KIN-1 at the L4 stage is lethal and animals do not reach adulthood (scale bar, 500 μm). Raw data underlying panels C, D, E, and F can be found in S6 Data.

*drl-1* mutant background (Fig 6B). Since FSHR-1 functions in the intestine to oppose DRL-1 signaling, we then performed intestine-specific RNAi against *gsa-1*, *acy-4*, and *kin-1* in the *drl-1* mutant and measured P*vit-3*::*mCherry* expression levels and body size (Fig 6C and 6D). Indeed, inactivation of this canonical PKA activation pathway in the intestine suppressed the *drl-1* mutation to levels similar to *fshr-1* knockdown. Notably, knockdown of *kin-1* by RNAi failed to strongly suppress the vitellogenesis and body size defects exhibited by the *drl-1* mutant. To investigate the role of KIN-1 in this pathway more rigorously, we introduced AID tags into two locations within the *kin-1* locus using CRISPR/Cas9 editing, resulting in AID tagging of all the *kin-1* isoforms. Depletion of KIN-1 in the intestine, but not in neurons, suppressed the vitellogenesis and body size defects resulting from *drl-1* knockdown (Fig 6E and 6F), which is consistent with our intestine-specific RNAi experiments. Interestingly, hypodermal depletion of KIN-1 at the L4 stage results in a highly penetrant bursting phenotype (Fig 6G), indicating that *kin-1* is required for developmental programs in the hypodermis or seam cells. While the ligand for FSHR-1 has remained elusive despite its widespread roles in immunity, stress responses, and germline development [34,37,38], our results are consistent with the possibility that FLR-2 is a ligand for FSHR-1 and induces cAMP signaling and PKA activation in the intestine. Indeed, FLR-2 was recently shown to bind to the extracellular domain of FSHR-1 in response to freeze-thaw stress [39], further supporting our conclusion that FLR-2 and FSHR-1 act in the same pathway.

## FLR-2 and DRL-1/FLR-4 inversely regulate p38 signaling to tune development

It is possible that FLR-2/FSHR-1/PKA signaling functions in parallel to DRL-1/FLR-4 to differentially regulate a core developmental pathway. Intriguingly, mutations in components of the p38 MAPK pathway, including *tir-1* (orthologue of the human TIR domain protein SARM1), *nsy-1* (MAPKKK), *sek-1* (MAPKK), or *pmk-1* (MAPK), suppresses the increased life span of *flr-4* and *drl-1* mutants [18,20]. The p38/PMK-1 pathway functions broadly in innate immunity [21,22], response to oxidative stress [23], development [40–42], and longevity [43]. Moreover, loss of *flr-4* or *drl-1* results in hyperphosphorylation of PMK-1 [18,20], which may promote slower developmental rates [41]. Consistent with these previous observations, our loss-of-function mutations in *drl-1* or *flr-4*, as well as intestinal depletion of AID::DRL-1 or AID::FLR-4 with auxin, increased the levels of active, phosphorylated PMK-1 (S9 Fig).

We hypothesized that hyperactivation of p38/PMK-1 may underlie the reduced developmental rates, small body sizes, and impaired vitellogenin production of the *drl-1* and *flr-4* mutants. Indeed, knockdown of the p38 pathway components *tir-1*, *nsy-1*, *sek-1*, or *pmk-1* by RNAi restored P*vit-3:mCherry* expression and increased the body size of *drl-1* mutant animals to varying degrees (S10A and S10B Fig). Moreover, intestine-specific knockdown of *pmk-1* partially suppressed the vitellogenesis defects, small body size, and slow growth of the *drl-1* mutant (Fig 7A–7C). Intestinal knockdown of *pmk-1* also partially suppressed the body size defects of animals depleted of intestinal FLR-4 (S10C Fig). Importantly, loss of *flr-2* was significantly more effective in suppressing DRL-1 depletion than a null mutation in either *pmk-1* or *tir-1* (Fig 7D), suggesting that FLR-2/FSHR-1 may function through multiple parallel pathways to modulate growth and lipid homeostasis.

Although our genetic data argue that FLR-2/FSHR-1 activates the p38 pathway, the site of this regulation is unknown. The MAP3K gain-of-function (gf) mutation, *nsy-1(ums8)*,

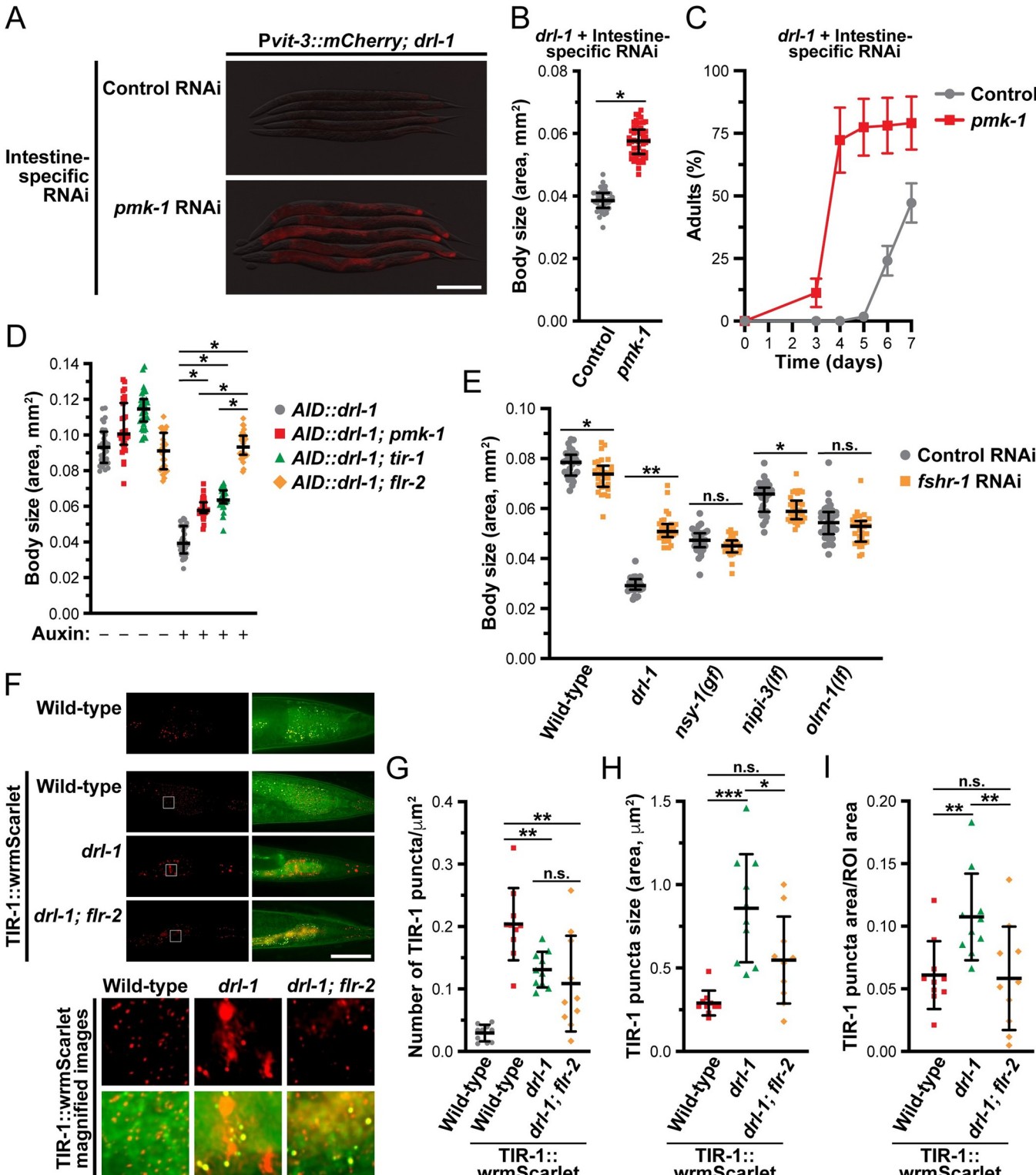

**Fig 7. DRL-1 and FLR-2/FSHR-1 signaling balance p38/PMK-1 activity to promote growth and development.** (**A**) Representative overlaid DIC and mCherry fluorescence images of day 1 adults (scale bar, 200 μm), (**B**) body size (day 1 adults; median and interquartile range; *, $P = 3 \times 10^{-36}$, $t$ test), and (**C**) growth rate (mean +/− SEM) of *drl-1(rhd109)* animals subjected to intestine-specific RNAi. (**D**) Body size of *AID::drl-1* day 1 adult animals following intestinal depletion of DRL-1 with 4 mM auxin (median and interquartile range; *, $P < 0.0001$, one-way ANOVA). (**E**) Body size of wild-type and the indicated mutants after control or *fshr-1* RNAi (median and interquartile range; *, $P < 0.01$, **, $P < 0.0001$, $t$ test). These mutations have been previously shown to activate p38/

PMK-1 signaling. (**F**) Representative images of TIR-1::wrmScarlet localization in posterior intestinal cells (scale bar, 50 μm). The red channel shows the TIR-1 puncta (left panels), the green/red overlay shows yellow puncta that result from intestinal autofluorescence (right panels), and wild-type animals lacking TIR-1:: wrmScarlet are included as a negative control (top panels). Magnifications of the boxed areas are displayed in the bottom images. Quantification of the (**G**) total number, (**H**) absolute size, and (**I**) relative size of TIR-1 puncta in day 1 adult wild-type and *drl-1(rhd109)* mutant animals (mean +/− SD; ns, not significant, *, $P < 0.05$, **, $P < 0.01$, ***, $P < 0.0001$, one-way ANOVA). Raw data underlying panels B, C, D, E, G, H, and I can be found in S7 Data.

hyperactivates downstream p38 signaling, resulting in developmental delay and small body size [41]. Using this mutant, we tested whether *fshr-1* functions genetically upstream of *nsy-1*. While *fshr-1* RNAi strongly suppresses the *drl-1(rhd109)* mutation, it fails to suppress the small body size of the *nsy-1* gf mutant (Fig 7E), indicating that FSHR-1 likely functions upstream of NSY-1 to regulate development. Moreover, other genetic perturbations that induce p38 hyperactivation include mutations in the neuronal developmental regulator *olrn-1* or the intestinal pseudokinase *nipi-3* [40,42,44]; however, knockdown of *fshr-1* failed to suppress the body size defects caused by the *olrn-1* or *nipi-3* mutations, indicating that these pathways function independently of *fshr-1* (Fig 7E).

Given that FSHR-1 functions upstream of NSY-1, we investigated whether the activity of TIR-1, the *C. elegans* orthologue of SARM1 (sterile alpha and TIR motif-containing 1) and upstream regulator of the p38 pathway [45,46], is modified by loss of *drl-1*. TIR-1 activation is triggered by a stress-induced phase transition that promotes protein oligomerization, $NAD^+$ glycohydrolase activity, and stimulation of the downstream NSY-1/SEK-1/PMK-1 pathway [24,47]. Using a strain expressing TIR-1::wrmScarlet [24], we tested whether loss of *drl-1* could enhance TIR-1 phase transition, which is visible as intestinal puncta that are distinct from the autofluorescent gut granules. While mutation of *drl-1* stimulated a reduction in the number of TIR-1::wrmScarlet puncta, the size of the puncta was markedly increased, suggesting that loss of *drl-1* likely induces TIR-1::wrmScarlet phase transition (Fig 7F–7I). This *drl-1*-induced oligomerization of TIR-1 was suppressed by loss of *flr-2*. Together, these data suggest that DRL-1/ FLR-4 and FLR-2/FSHR-1 exert opposing effects on TIR-1 to govern the activity of p38 signaling and animal development.

Our results suggest that FLR-2/FSHR-1 signaling stimulates p38 activity in the absence of *drl-1/flr-4*; however, it remains unclear whether PKA mediates these effects downstream of FLR-2. A gain-of-function mutation in Gα$_s$, *gsa-1(ce81)*, or a loss-of-function mutation in the inhibitory regulator of PKA, *kin-2(ce179)*, hyperactivates PKA signaling [48] and reduces body size. This body size defect is partially suppressed by *pmk-1* RNAi (S11A Fig), suggesting that PKA functions either upstream or in parallel to PMK-1. To directly test whether PKA activation stimulates p38, we measured TIR-1 oligomerization and PMK-1 phosphorylation in the PKA hyperactivation mutants, finding that independent activation of PKA failed to induce TIR-1 phase transition or PMK-1 phosphorylation (S11B–S11D Fig). While it is possible that PKA only acts on the p38 pathway when *drl-1* or *flr-4* is lost, it is more likely that FLR-2 stimulates a PKA-independent parallel pathway, possibly mediated by an alternative G protein signaling cascade, to modulate the activity of TIR-1.

While the transcription factors that act downstream of p38 signaling to regulate stress responses have been well studied in *C. elegans* [49], those that function in p38-regulated development are not well understood. Thus, we performed a small-scale RNAi screen to identify transcriptional regulators that function downstream of *drl-1* to regulate development. Using the P*vit-3*::*mCherry* reporter, we first assessed whether knockdown of each transcription factor could suppress the vitellogenesis defects of the *drl-1* mutant. Interestingly, knockdown of *pha-4*, a FoxA transcription factor, partially suppressed the vitellogenesis and body size defects conferred by the *drl-1* mutation (Fig 8A and 8B). PHA-4 functions broadly in *C. elegans* development but also has a distinct role in promoting longevity in response to dietary restriction [50].

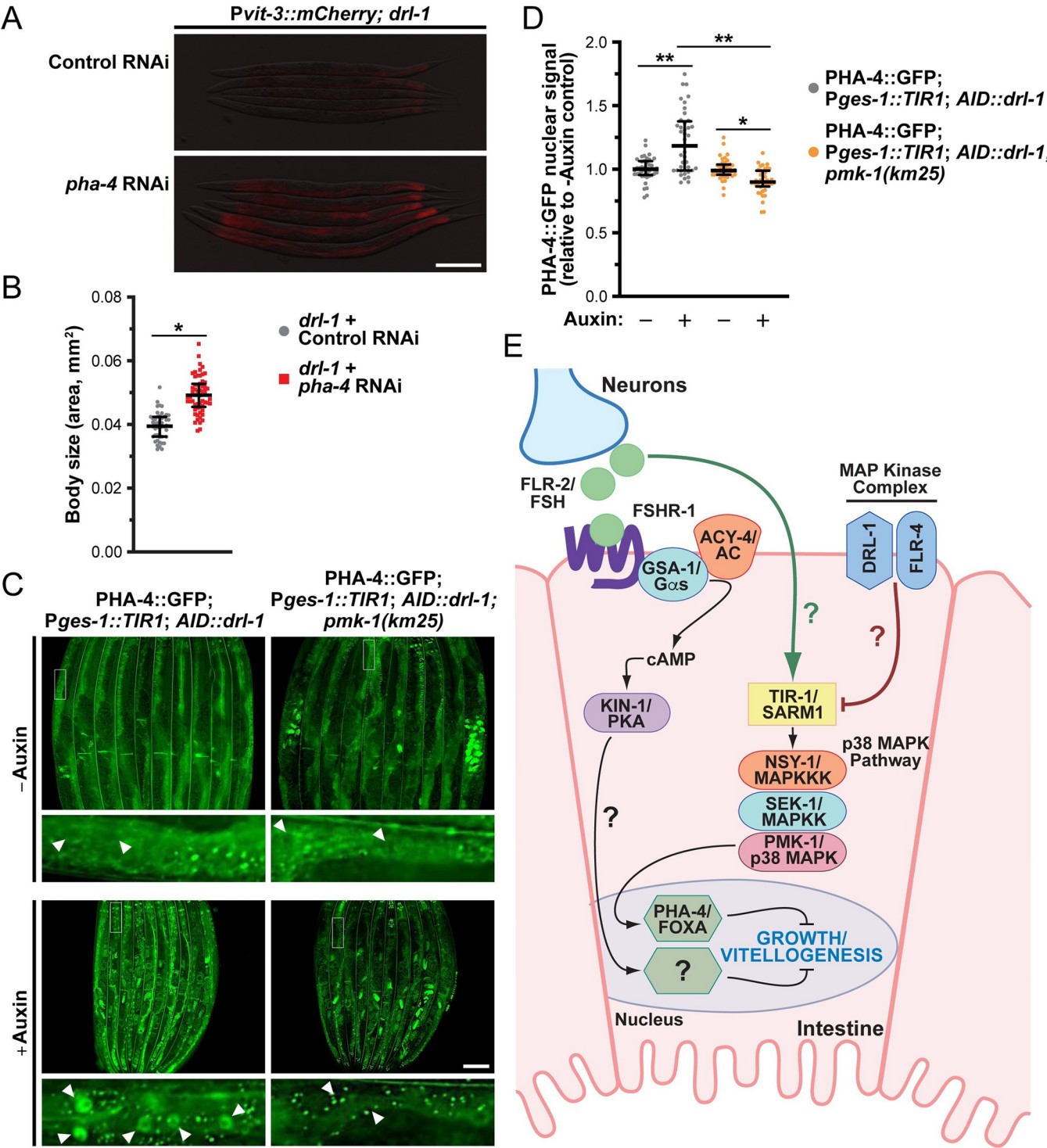

**Fig 8. PHA-4/FOXA suppresses growth upon loss of *drl-1*.** (**A**) Representative fluorescence images of P*vit-3::mCherry* reporter expression (F1 animals; scale bar, 200 μm) and (**B**) body size (P0 animals; median and interquartile range; \*, $P = 7 \times 10^{-17}$, t test) of *drl-1(rhd109)* day 1 adults after knockdown of *pha-4* by RNAi. (**B**) This control RNAi data is also presented in Fig 5E. (**C**) GFP fluorescence images (magnifications of the outlined regions are shown below the primary image with white arrowheads indicating intestinal nuclei; scale bar, 100 μm) and (**D**) quantification (median and interquartile range; \*, $P < 0.05$, \*\*, $P < 0.0001$, one-way ANOVA) of PHA-4::GFP nuclear localization after intestine-specific depletion of AID::DRL-1 using 4 mM auxin in wild-type or *pmk-1 (km25)* animals reared on *E. coli* OP50. (**E**) A model illustrating the opposing effects of DRL-1/FLR-4 and FLR-2/FSHR-1 on p38-mediated growth. Raw data underlying panels B and D can be found in S8 Data.

Furthermore, *pha-4* is required for the life span increase that results from loss of *drl-1* or *flr-4* [17,20]. Thus, we reasoned that DRL-1/FLR-4 may impact the localization of PHA-4, which we assessed using a strain that expresses an endogenously tagged PHA-4::GFP to avoid potential artifacts of PHA-4 overexpression. Indeed, depletion of DRL-1 stimulated the accumulation of PHA-4::GFP protein in the nucleus of intestinal cells (Fig 8C and 8D). Moreover, the PHA-4 nuclear accumulation was dependent on *pmk-1* regardless of the food source, suggesting that DRL-1, and downstream p38/PMK-1 signaling, may regulate the nuclear translocation of PHA-4/FOXA (Figs 8C, 8D, S12A and S12B). It is possible that PMK-1 promotes the nuclear accumulation of PHA-4, where it directly represses the expression of specific developmental genes. In support of this hypothesis, PHA-4 is found at the promoters of the *vit* genes at low levels specifically at adulthood (S12C Fig), possibly mediated by the basal activity of PMK-1. Dynamic nuclear translocation of PHA-4 has not yet been observed in *C. elegans* [50,51]; however, these previous studies used PHA-4 overexpression transgenes, while our experiments employed a CRISPR/Cas9-based GFP knock-in at the *pha-4* locus, which tags all isoforms. Notably, the human FOXA proteins show different abilities to dynamically shuttle between the nucleus and cytoplasm [52]. Together, our data demonstrate that DRL-1 signaling governs TIR-1 phase transition to module p38 activity and PHA-4 localization in the intestine to tune the developmental rate of *C. elegans*.

## Discussion

Here, we demonstrate that the DRL-1 and FLR-4 MAP kinases play crucial roles in managing organismal development, growth, and lipid homeostasis. The action of these pro-growth MAPKs, which form a protein complex on the plasma membrane of intestinal cells, is opposed by the neurohormone FLR-2, the *C. elegans* orthologue of the follicle-stimulating hormone (FSH), and its putative intestinal G protein–coupled receptor FSHR-1 and downstream $G\alpha_s$/cAMP/PKA signaling (Fig 8E). In turn, MAPK and FSH-like signaling balance the activity of p38/PMK-1 by modulating the oligomerization of TIR-1/SARM1, thereby tuning developmental rate and growth. At least some of the intestinal response to altered p38 activity is conferred at the level of transcription, as loss of *pha-4* partially rescues the developmental defects displayed by the *drl-1* mutant. Our study establishes a new mechanism by which the intestinal p38 signaling pathway integrates tissue-specific signals to govern organismal growth and development.

Our work suggests that loss of either *drl-1* or *flr-4* results in widespread misuse of energy resources, causing slow growth, small body size, and loss of fat storage. It is likely that impaired vitellogenin production is a consequence of this dietary restriction-like metabolic state and not due to impaired developmental programs, as other models of dietary restriction also display reduced vitellogenin levels [10]. Consistently, post-developmental depletion of DRL-1 or FLR-4 reduced *vit-3* reporter expression and decreased body size. Notably, mutations that abrogate the presumptive kinase activity of DRL-1 or FLR-4 result in strong loss-of-function phenotypes, suggesting that both proteins may function in canonical MAP kinase signaling in *C. elegans*. While the kinase domain of DRL-1 shares highest similarity to the mammalian MEKK3 protein [17,30], the FLR-4 kinase domain is most similar to the catalytic domains of the GCKIII kinases (i.e., STK24/MST3, STK25/SOK1, and STK26/MST4) [19]. Interestingly, knockdown of STK25 increases β-oxidation and impairs lipid accumulation in human hepatocytes [53], which is consistent with *flr-4* loss-of-function phenotypes.

Here, we show that DRL-1 and FLR-4 form a protein complex. While MEKK3 and GCKIII kinases are not known to directly interact, both have been shown to interact with the CCM (cerebral cavernous malformation) adaptor proteins [54]. Specifically, CCM2/OSM scaffolds

MEKK3 at membranes where it activates p38 during osmotic stress [55,56]. In this cellular context, MEKK3 promotes p38 activity; however, in different cell types, CCM2 and CCM3 have been shown to have no impact on p38 activity or to negatively regulate p38 [57,58], suggesting that cell type-specific scaffolds may be crucial to defining how the MEKK3 or the GCKIII kinases impact p38 activity. In *C. elegans*, the CCM orthologues KRI-1, CCM-2, and CCM-3 have poorly defined roles in intestinal metabolism and additional studies will be crucial to establishing whether DRL-1 and FLR-4 function in complex with these adaptor proteins.

While previous studies have established a genetic interaction between *drl-1/flr-4* and the p38 pathway [18,20], the molecular mechanism of this crosstalk has been unclear. We demonstrate a surprising mode of regulation whereby oligomerization of TIR-1/SARM1 is repressed by DRL-1, and likely FLR-4 as well, to modulate downstream p38 signaling. Conversely, FSH-like signaling promotes TIR-1 phase transition. Notably, previous work has demonstrated that activation of TIR-1 by pathogen or nutrient stress stimulates an increase in TIR-1 puncta number [24]; however, we observed a modest decrease in the number of TIR-1 puncta and a dramatic increase in puncta size. It is possible that our imaging and analysis approach, using a different microscopy setup and deconvolution/denoising software, accounts for this difference. Alternatively, loss of *drl-1* may impact TIR-1 oligomerization differently than pathogen or nutrient stress. Nonetheless, we find that robust TIR-1 oligomerization correlates with activation p38/PMK-1 signaling, which is consistent with the previous findings [24].

Regulation of TIR-1/SARM1 phase transition via phosphorylation has not been demonstrated; however, in human cells, phosphorylation of SARM1 by JNK promotes the intrinsic NAD$^+$ hydrolase activity of the TIR domain [59]. It is possible that DRL-1 or FLR-4 may directly phosphorylate TIR-1 to prevent phase transition and restrict NAD$^+$ hydrolase activity. This event may be opposed by FLR-2/FSHR-1 signaling; however, this is unlikely to require PKA, as loss of *kin-2* does not stimulate TIR-1 oligomerization. Identification of the targets of DRL-1/FLR-4 will be crucial to untangling these regulatory mechanisms. Intriguingly, in cardiomyocytes, STK25 inhibits PKA via phosphorylation of the regulatory subunit PRKAR1A [60], suggesting that FLR-4 may function in a similar manner to regulate PKA in *C. elegans*.

While direct regulation of TIR-1 via phosphorylation is plausible, it is also possible that DRL-1/FLR-4 and FSH-like signaling act indirectly to control p38/PMK-1 signaling. A recent study demonstrated that cholesterol deficiency or loss of the NHR-8 nuclear hormone receptor, which is required for maintaining cholesterol homeostasis in *C. elegans*, stimulates p38 activity by promoting TIR-1 phase transition and NAD$^+$ hydrolase activity [24,61]. Thus, loss of DRL-1 or FLR-4 could induce similar conditions of cholesterol mishandling, which would be consistent with the impaired intestinal lipid homeostasis, the altered cytoprotective responses, and heightened detoxification response that is observed in *drl-1* mutant animals [17,30]. Notably, loss of *drl-1* up-regulates numerous cytochrome P450 and UDP-glucuronosyltransferase detoxification genes, which could deplete cholesterol stores by increasing flux through sterol modification and catabolic pathways [17,62].

We demonstrate that the opposing action of FSH-like signaling and DRL-1/FLR-4 on p38/PMK-1 signaling governs the nuclear localization of PHA-4/FOXA in the intestine. To our knowledge, this is the first demonstration that PHA-4/FOXA nuclear localization is controlled by p38 signaling. The mammalian FOXA transcription factors are crucial regulators of early development and post-natal metabolic homeostasis [63]. Similarly, in *C. elegans*, PHA-4 specifies pharyngeal cell fates and is required for the development of the foregut [64–66], as well as post-embryonic regulation of metabolism and aging [50,51,67]. Given the well-defined role of PHA-4 in promoting development, it is surprising that PHA-4 restricts, either directly or indirectly, vitellogenin production and body size upon loss of intestinal DRL-1. It is possible that

phosphorylation by PMK-1 directs not only the nuclear localization of PHA-4 but also its preference for transcriptional targets. Moreover, it will be crucial to define the tissue of action and the developmental role of PHA-4 during dietary restriction (DR), as *pha-4* is required for DR-induced longevity [50].

We found that *flr-2* mutations are stronger suppressors of *drl-1* mutant phenotypes than the *pmk-1* null mutation, suggesting that FSH-like signaling controls a p38-independent pathway. Consistently, intestinal FSHR-1 acts in parallel to the p38 and insulin signaling pathways to support the innate immune response [37]. Moreover, the developmental defects, metabolic reprogramming, and misexpression of several cytoprotective genes induced by loss of *drl-1* are not strictly dependent on p38 signaling [18,30]. Thus, it is likely that the FLR-2/FSHR-1/PKA signaling pathway also promotes metabolic and developmental defects in *drl-1* mutants through a p38-independent mechanism. Notably, PKA promotes lipid mobilization in response to fasting and cold stress, likely by phosphorylating and stabilizing the ATGL-1 lipase on the surface of intestinal lipid droplets [68–70]. Similar to the fasting response, knockdown of *drl-1* stimulates β-oxidation and up-regulation of lipid catabolism genes (e.g., *cpt-3*/carnitine palmitoyl transferase), which, in turn, likely increases energy production through mitochondrial oxidative phosphorylation [17,69]. Thus, it is possible that the p38-independent metabolic defects displayed by the *drl-1* mutant are a result of PKA-dependent lipid mishandling of intestinal lipid droplets. In the future, it will be crucial to define the metabolic role of FSH signaling in mammalian non-reproductive tissues, including the liver and adipose tissue, as well as assess whether MAPK signaling impinges on these pathways to integrate metabolic and developmental programs.

The DRL-1 and FLR-4 MAPKs are crucial to maintaining cellular homeostasis and balancing pro-growth programs against energy utilization. We propose that FSH-like signaling and DRL-1/FLR-4 are likely to be dynamically regulated by nutritional inputs or environmental stimuli. Conceivably, the metabolic reprogramming and life span extension triggered by dietary restriction, or the elevated lipid utilization in response to short-term fasting, could engage neuronal FSH-like signaling to promote breakdown of intestinal lipids. Intriguingly, an *E. coli* HT115 diet further impairs the developmental rate of the *drl-1* mutant, and loss of *flr-2* fails to suppress this defect, suggesting that additional nutritional or sensing pathways may function in parallel with FSH-like signaling to restrict development [33]. This work establishes the framework for identifying these unknown regulators, which will be vital to gaining a holistic view of the homeostatic mechanisms that promote development and reproductive fitness.

## Materials and methods

### *C. elegans* strains

*C. elegans* strains were cultured on NGM media seeded with *E. coli* OP50 or HT115(DE3) [71]. Animals were reared at 20˚C unless specified otherwise. For auxin-inducible degradation experiments, embryos were transferred to plates containing 4 mM naphthaleneacetic acid (K-NAA, PhytoTech) and grown to adulthood. All strains used in this study are listed in S1 File.

### Generation and imaging of transgenic animals

Strains carrying the high-copy *mgIs70[Pvit-3::GFP]* transgene or the single-copy *rhdSi42[Pvit-3::mCherry]* transgene have been previously described [14]. The *drl-1* rescue constructs were generated by fusing the *col-10* promoter (chromosome V: 9,166,416–9,165,291; WS284) or the *vha-6* promoter (chromosome II: 11,439,355–11,438,422; WS284) to the *drl-1* cDNA (1,770 bp of coding sequence with 141 bp of 3′ UTR) via Gibson assembly to generate plasmids pRD141 and pRD142, respectively [14,72]. The resulting plasmids were microinjected into *drl-1*

*(rhd109); mgIs70[Pvit-3::GFP]* animals at 20 ng/µl, along with 2.5 ng/µl pCFJ90(P*myo-2*::*mCherry*) and 77.5 ng/µl of 2-Log DNA ladder (New England BioLabs), to generate 2 independent strains expressing *Ex[Pcol-10::mCherry::his-58::SL2::drl-1 cDNA]* (DLS515, DLS516) and *Ex[Pvha-6::mCherry::his-58::SL2::drl-1 cDNA]* (DLS513, DLS514). For the pan-neuronal *flr-2* rescue transgene, the *sng-1* promoter (chromosome X: 7,325,641–7,327,607; WS284) was fused to the *flr-2* gene (including the 3′ UTR) via Gibson assembly to generate the pRD157 plasmid. This plasmid, which is derived from the MosSCI-compatible pCFJ151 plasmid, was microinjected into EG6699 to generate the single-copy integrant *rhdSi46[Psng-1::mCherry::his-58::SL2::flr-2]* as previously described [73]. All strains carrying *mgIs70* or *rhdSi42* were imaged on a Nikon SMZ-18 Stereo microscope equipped with a DS-Qi2 monochrome camera.

## CRISPR/Cas9 gene editing

Generation of the *drl-1* deletion alleles (*rhd109* and *rhd110*) were generated using the *pha-1* co-conversion approach, as previous described [74]. All additional edits were performed by microinjection of Cas9::crRNA:tracrRNA complexes (Integrated DNA Technologies) into the germlines of *C. elegans* animals as previously described [75]. Large dsDNA donor molecules with approximately 40 bp homology arms on each end were prepared by PCR using Q5 DNA Polymerase (New England BioLabs) and purified using HighPrep PCR Clean-up beads (MagBio) per the manufacturers' instructions. The DNA repair templates were melted and reannealed prior to microinjection [75]. The PCR templates used to generate the dsDNA donor molecules were pRD156, pBluescript II(*linker::mKate2::TEV::linker::3xFLAG::AID*); pRD160, pBluescript II(*AviTag::linker::2xTEV::linker::3xHA::linker::mNeonGreen::TEV::linker::3x-FLAG::AID*); and pRD174, pBluescript II(*AviTag::linker::2xTEV::linker::3xHA::linker::mGreen-Lantern::TEV::linker::3xFLAG::AID*). To generate missense mutations using CRISPR/Cas9 gene editing, single-stranded oligodeoxynucleotides were used as donor molecules [75]. All CRISPR crRNA guide sequences are listed in S1 File.

## RNAi experiments

*E. coli* HT115(DE3) strains carrying the control L4440 plasmid or individual RNAi plasmids for gene expression knockdown were grown for approximately 16 hours in Luria–Bertani medium containing ampicillin (50 µg/ml), concentrated by 20 to 30× via centrifugation, and seeded on NGM plates containing 5 mM isopropyl-β-D-thiogalactoside (IPTG) and 50 µg/ml ampicillin. Plates seeded with RNAi bacteria were maintained at room temperature overnight to induce expression of the dsRNAs. All RNAi clones were selected from the Ahringer or Ahringer Supplemental RNAi library and confirmed by Sanger sequencing, with the exception of *nsy-1*, which has been described previously [41], and *sek-1* and *pmk-1*, which were generated by cloning cDNA fragments into the pL4440 vector using standard techniques. We selected 4 different *kin-1* RNAi clones from the Ahringer library to initially test for their ability to suppress the *drl-1* mutant phenotypes. Three of these RNAi clones caused lethality or severe developmental phenotypes, while the weakest clone permitted growth to adulthood but produced modest *drl-1* suppression phenotypes, likely due to incomplete knockdown of the *kin-1* transcript. Synchronized *C. elegans* L1 larvae were generated by bleaching gravid animals to liberate embryos followed by overnight incubation in M9 buffer. Synchronized L1s were dropped on RNAi plates, grown at 20°C until they were day 1 adults (72 to 120 hours), and processed for imaging.

## Growth, life span, and brood size assays

Animals were grown on their respective *E. coli* food sources (OP50 or HT115) for at least 2 generations without starvation prior to being assayed growth, life span, and brood size. For

developmental growth rate assays, 100 to 200 eggs were picked from plates with freshly laid embryos (16 hours or less) to new plates. Animals were scored every 24 hours for 7 days for the presence of gravid adults, which were promptly removed and recorded as having reached adulthood. For RNAi-treated animals, synchronized L1s were dropped on RNAi plates grown for up to 7 days and similarly scored. For developmental experiments employing auxin-inducible degradation of AID::DRL-1 or AID::FLR-4, freshly laid embryos were picked to plates containing 4 mM naphthaleneacetic acid and worm growth was scored as described above.

For body size measurements, L4 animals were picked to fresh plates and imaged 24 hours later on a Nikon SMZ-18 Stereo microscope equipped with a DS-Qi2 monochrome camera. Using the Fiji software [76], animals were outlined by hand and the number of pixels were measured, which was converted to a square micron value based on the known imaging settings. Body size data in $mm^2$ are presented as the median with the interquartile range. A one-way ANOVA with a Bonferroni correction for multiple testing was performed to determine whether samples were significantly different.

Life span assays were performed at 25˚C in the presence of 50 μM FUDR unless otherwise noted. Briefly, well-fed L4 animals were picked to FUDR-containing plates and animals (30 animals/plate, 120 to 150 total animals) were maintained on uncontaminated plates and scored daily for survival as previously described [16]. A log-rank test was applied to determine whether survival curves were significantly different.

For brood sizes experiments, L4 animals (N2, $n = 11$; *drl-1(rhd109)*, $n = 23$; *drl-1(rhd109); flr-2(rhd117)*, $n = 11$) were picked to individual plates and transferred daily throughout the reproductive period. Progeny were counted as L3 or L4 animals and data are reported as the mean +/− the standard deviation.

## Oil Red O staining

Approximately 75 L4 animals were picked to new plates, harvested 24 hours later as day 1 adults in S buffer, washed, fixed with 60% isopropanol, and stained for 7 hours with 0.3% Oil Red O as previously described [16]. Animals were mounted on 2% agarose pads and imaged with a Nikon SMZ-18 Stereo microscope equipped with a DS-Qi2 monochrome camera for intensity analyses or a Nikon Ti2 widefield microscope equipped with a DS-Fi3 for representative color images. Quantification of Oil Red O staining was performed in ImageJ by manually outlining worms and determining the mean gray value in the worm area. Each value was subtracted from 65,536, the maximum gray value for 16-bit images. The data were plotted using Prism 9 and a one-way ANOVA with a Bonferroni correction for multiple testing was performed to calculate *P* values.

## Quantitative PCR

L1 animals were synchronized by bleaching and grown to the first day of adulthood prior to being harvested in M9 buffer, washed, and flash frozen. The total RNA was isolated using Trizol (Thermo Fisher), followed by chloroform extraction and precipitation with isopropanol. DNase treatment, followed by cDNA synthesis with oligo(dT) priming, was performed using the SuperScript IV VILO Master Mix with ezDNase Kit according to the manufacturer's instructions (Thermo Fisher). Quantitative PCR was performed exactly as previously described [16]. All primer sequences are listed in S1 File. Expression data are presented as the mean fold change relative to a control (N2 for mutant analyses, L4440 for RNAi analyses) with the standard error of the mean (SEM) reported for 3 independent experiments.

## Western blot analyses

Animals were synchronized by bleaching and the resulting L1s were grown to day 1 adults. The animals were harvested in M9 buffer, washed 3 times, and the worm pellets were snap frozen in liquid nitrogen. The pellets were resuspended in an equal volume of 2× RIPA buffer (Cell Signaling Technology) containing a 2× Halt Protease and Phosphatase Inhibitor Cocktail (Thermo Fisher) and homogenized for approximately 15 seconds with disposable pellet pestle (Thermo Fisher) mounted in a cordless drill (Dewalt). Samples (approximately 200 μL) were then subjected to sonication (30-second on/off cycles, 10 cycles) using a Bioruptor Pico sonication instrument (Diagenode). Whole cell lysates were cleared by centrifugation and protein concentrations were determined using the DC Protein Assay (BioRad) according to the manufacturer's instructions. Equal amounts of protein for each sample (approximately 50 μg) were resolved by SDS-PAGE, transferred to a PVDF membrane, blocked in 5% nonfat dry milk (BioRad), and probed with either anti-phospho-p38 MAPK (Thr180/Tyr182; #9211, Cell Signaling Technology), anti-PMK-1 [77], or anti-Actin antibodies (ab3280, Abcam).

For co-immunoprecipitation (co-IP) experiments, approximately 50,000 day 1 adults were harvested, homogenized, and resuspended in lysis buffer exactly as previously described [16]. Worm lysates were sonicated with the Bioruptor Pico sonication instrument (Diagenode) and cleared by centrifugation. For co-IP of mKate2::3xFLAG::DRL-1 with 3xHA::mGL::FLR-4 from strain DLS781, the lysate was split in 3 equal parts and subjected to a mock IP, anti-HA IP (2 μg of 3F10; 11867423001, Sigma), or anti-FLAG IP (5 μg of M2; F1804, Sigma). Prior to the immunoprecipitations antibodies were bound to Protein G Dynabeads (Invitrogen) with the mock IP sample lacking antibody. Western blotting of immunoprecipitated proteins was performed as described above using anti-FLAG (F1804, Sigma) or anti-HA (11867423001, Sigma) antibodies.

## EMS mutagenesis and identification of causative mutations

Mutagenesis of *drl-1(rhd109); mgIs70* (DLS364) animals with ethyl methanesulfonate (EMS, Sigma-Aldrich) was performed exactly as previously described [14]. A total of 50,000 haploid genomes were screened across 6 genetically distinct pools. Suppressor mutants (approximately 130 viable strains) were selected based on increased growth rate and/or *mgIs70 [Pvit-3::GFP]* expression relative to the parental strain.

Suppressor strains were selected and backcrossed to the DLS364 strain 2 times. F2 recombinants from the second backcross that displayed the suppression phenotype were singled, the resulting plates were allowed to starve, the animals across all individual plates were pooled, and genomic DNA was prepared from the pool using the Qiagen Gentra Puregene Tissue Kit [78]. Whole genome sequencing libraries were prepared using the TruSeq DNA PCR-Free kit (Illumina) and sequenced on an Illumina HiSeq 4000 instrument according to the manufacturer's instructions. Identification of candidate suppressor mutations was performed as previously described [79] using in-house scripts.

## Reporter imaging and quantification

To measure P*vit-3*::*mCherry* fluorescence, strains carrying the *rhdSi42* reporter were grown to the L4 stage, picked to new plates, mounted 24 hours later as day 1 adults with 25 mM levamisole on a 2% agarose pad, and imaged with a Nikon SMZ-18 Stereo microscope equipped with a DS-Qi2 monochrome camera. Worm bodies were traced in the brightfield channel and mean intensities (gray values) were calculated in the mCherry channel using ImageJ. The data were plotted in Prism 9 and a one-way ANOVA with a Bonferroni correction was performed to calculate *P* values.

For TIR-1::wrmScarlet imaging, animals were reared on *glo-3* RNAi for several generations using the RNAi-competent *E. coli* OP50(xu363) strain [80]. Knockdown of *glo-3* reduces intestinal autofluorescence and facilitates TIR-1 imaging [24]. At the L4 stage, animals were picked to fresh *glo-3* RNAi plates, mounted on a 2% agarose pad 24 hours later, and imaged with a CFI Apo 60X Oil TIRF objective on a Nikon Ti2 widefield microscope equipped with a Hamamatsu ORCA-Fusion BT camera. The *mKate2::drl-1*; *mGreenLantern::flr-4; glo-4(ok623)* CRISPR knock-in animals were imaged at 60×, and animals expressing the TIR1::F2A:: mTagBFP2::AID::NLS transgene were imaged at 20× on the Nikon Ti2 microscope.

TIR-1 puncta were measured in the posterior intestine since these 2 to 4 epithelial cells are not obstructed by other tissues. Z-stacks were cropped to a depth of 3 to 4 μm to capture planes where the intestine can be clearly visualized. All Z-stacks were denoised, deconvoluted, and compressed into a single image using the Nikon NIS-Elements analysis software. ROIs were hand drawn around the 2 to 4 posterior-most intestinal cells and the TIR-1 puncta were quantified using the object counts feature in NIS-Elements, which separated TIR-1::wrmScarlet puncta from the autofluorescent gut granules that appear in both the FITC and mCherry channels. TIR-1 puncta counts, puncta area, and puncta area relative to the ROI area were plotted in Prism 9, and a one-way ANOVA with a Bonferroni correction was performed to calculate *P* values.

For PHA-4::EGFP imaging, *reSi5; mKate2::3xFLAG::AID::drl-1; pha-4::EGFP* animals were reared on OP50(xu363) or HT115(DE3) *glo-3* RNAi bacteria in the presence of 4 mM auxin for multiple generations to reduce gut autofluorescence and deplete the DRL-1 protein. L4 animals were picked to new plates and imaged 24 hours later as day 1 adults with a 10× objective on the Nikon Ti2 microscope. All Z-stacks were denoised and deconvoluted using the Nikon NIS-Elements analysis software and a single plane was selected for analysis based upon which had the best resolved intestinal nuclei. The nuclei of the 2 visible anterior and posterior-most intestinal cells were outlined by hand in ImageJ and GFP fluorescent signal intensity measurements were obtained by recording the mean gray value of each nucleus. The mean intensity values for each anterior nucleus for the auxin-treated animals (approximately 10 individuals/ sample) were divided by the mean intensity value of all the anterior nuclei for the minus auxin control animals. An identical analysis was performed for the posterior nuclei, and the data were plotted in Prism 9 and a one-way ANOVA with a Bonferroni correction was performed.

## Supporting information

**S1 Fig. Intestinal DRL-1 controls vitellogenin production and growth.** (**A**, **B**) Representative fluorescence and DIC overlaid images of *mgIs70[Pvit-3::GFP]* reporter expression in various *drl-1* mutants (scale bar, 200 μm). The *mgIs70* transgene is a high-copy transgene. (**C**) Intestinal, but not hypodermal, rescue of *drl-1(rhd109)* mutants with *drl-1* cDNA restores P*vit-3::GFP* expression (scale bar, 200 μm). (**D**) Growth rate (mean +/− SEM) and (**E**) body size (day 1 adults; median and interquartile range; *, *P* < 0.0001, one-way ANOVA) of wild-type and *drl-1* mutant animals. Raw data underlying panels D and E can be found in S9 Data. (TIF)

**S2 Fig. TIR1 expression from the P*col-10*::TIR1 transgene is high at the L4 stage but persists into adulthood.** (**A**) Representative BFP fluorescence images of L4s and day 1 adult animals carrying the *reSi1[Pcol-10::TIR1::F2A::mTagBFP2::AID\*::NLS::tbb-2 3'UTR]* transgene grown in the absence or presence of 4 mM auxin (scale bar, 100 μm). (**B**) Day 1 adult *reSi1 [Pcol-10::TIR1]; kin-1::AID* animals treated with or without 4 mM auxin and scored 24 hours later for motility defects and/or death (*n* = 50 animals per condition). (**C**) Development of *AID::drl-1* animals to the L4 stage (48 hours at 20°C) after hypodermal (*reSi1*) or intestinal

(*reSi5*) depletion of DRL-1 with 4 mM auxin (*n* = approximately 100 per condition). Raw data underlying panels B and C can be found in S10 Data.
(TIF)

**S3 Fig. *drl-1* and *flr-4* mutants display similar vitellogenesis and growth defects.** (**A**) Representative overlaid DIC and GFP fluorescence images of day 1 adult wild-type and mutant animals reared at 20 or 25˚C (scale bar, 200 μm). (**B**) Growth rate (mean +/− SEM) and (**C**) body size (day 1 adults; median and interquartile range; *, *P* < 0.0001, one-way ANOVA) of wild-type, *drl-1(rhd109)*, and *flr-4(ut7)* animals. (**D**) The percentage of animals at the L4 stage 48 hours after dropping eggs (white bars) or synchronized L1s (black bars) grown at 20˚C. Animals lacking *drl-1* (*rhd109* allele) or overexpressing *drl-1* (*rhdEx100* transgenics) were subjected to control, *drl-1*, or *flr-4* RNAi before scoring. Raw data underlying panels B, C, and D can be found in S11 Data.
(TIF)

**S4 Fig. *drl-1* and *flr-4* function in a cell-autonomous manner to regulate vitellogenin production.** (**A**) Representative fluorescence images of P*vit-3*::*mCherry* reporter expression in day 1 adult animals after whole-body or tissue-specific knockdown of *drl-1* or *flr-4* by RNAi (scale bar, 200 μm). Images of P*vit-3*::*mCherry* expression after adult-specific, intestinal depletion of (**B**) *mKate2*::*3xFLAG*::*AID*::*drl-1* or (**C**) *mNG*::*3xFLAG*::*AID*::*flr-4* with 4 mM auxin. (B, C) Auxin was applied to L4s for 24 hours (top), day 1 adults for 24 hours (middle), or day 1 adults for 48 hours (bottom) prior to imaging alongside of the no auxin controls (scale bars, 200 μm). Body size measurements of (**D**) wild-type, (**E**) *AID*::*drl-1*, and (**F**) *AID*::*flr-4* animals treated with 4 mM auxin as L4s and imaged as day 1 adults (left, D1Ad) or treated as day 1 adults and imaged as day 2 adults (right, D2Ad). For (**E**, **F**), whole life auxin treatments are also included. (**D**-**F**) All data are plotted as the median and interquartile range (n.s., not significant, *, *P* < 0.0001, one-way ANOVA). (**B**-**F**) All animals carry the P*ges-1*::*TIR1* transgene. Raw data underlying panels D, E, and F can be found in S12 Data.
(TIF)

**S5 Fig. Intestinal depletion of DRL-1 or FLR-4 paired with an *E. coli* HT115 diet results in severe growth defects and an extended life span.** (**A**) Growth rate (mean +/− SEM) of *mKate2*::*3xFLAG*::*AID*::*drl-1* animals with or without 4 mM auxin reared on *E. coli* OP50 or HT115. Longitudinal life span assays of (**B**) *mKate2*::*3xFLAG*::*AID*::*drl-1* or (**C**) *mNG*::*3xFLAG*::*AID*::*flr-4* animals grown at 20˚C with FUDR on *E. coli* OP50 or HT115 (P*ges-1*::*TIR1*, intestinal depletion; P*col-10*::*TIR1*, hypodermal depletion; P*rgef-1*::*TIR1*, pan-neuronal depletion). Control animals only carry the *TIR1* transgenes, and all strains were reared on 4 mM auxin from hatching. Raw data underlying panels A, B, and C can be found in S13 Data.
(TIF)

**S6 Fig. *flr-2* functions in a non-cell-autonomous manner to suppress the *drl-1(rhd109)* mutation.** (**A**) Overlaid DIC and GFP fluorescence images of day 1 adult wild-type, *drl-1 (rhd109)*, and *drl-1(rhd109)* double mutant animals (scale bar, 200 μm). (**B**) Representative images of day 1 adult wild-type, *flr-2(rhd117)*, *drl-1(rhd109)*, and *drl-1(rhd109); flr-2(rhd117)* animals stained with Oil Red O (scale bar, 200 μm). (**C**) Growth rate (mean +/− SEM) and (**D**) body size (day 1 adults) of wild-type, *drl-1(rhd109)* single and double mutants, and *flr-2* pan-neuronal rescue animals (P*sng-1*::*flr-2* is a single-copy rescue transgene). (**D**) Body size data are presented as the median and interquartile range (***, *P* < 0.0001, **, *P* = 0.0003, *, *P* = 0.04, one-way ANOVA). Wild-type animals have a significantly larger body size compared to all other strains (*P* < 0.0001). Raw data underlying panels C and D can be found in S14

Data.
(TIF)

**S7 Fig.** *flr-2* **functions in parallel to** *drl-1*. (**A**) A developmental time course of 3xHA::FLR-2 protein expression determined by western blotting of whole worm extracts. (**B**) Protein levels of 3xHA::FLR-2 after knockdown of *drl-1* or *flr-4* by RNAi. (**B**, **C**) The HA tag is inserted directly downstream of the signal peptide cleavage site and the faint upper bands in the HA blots are likely unprocessed FLR-2 protein. Western blot experiments were performed twice with similar results. (**C**) Body size data showing that overexpression of *flr-2* (P*sng-1*::*flr-2*) only modestly suppresses the *drl-1(rhd109)* mutation (day 1 adults, ***, $P < 0.0001$, *, $P = 0.0012$, one-way ANOVA). Raw data underlying panel C can be found in S15 Data, and raw images for panels A and B can be found in S2 and S3 Raw Images, respectively.
(TIF)

**S8 Fig.** *fshr-1* **mutations suppress the vitellogenin expression defects displayed by the** *drl-1 (rhd109)* **mutant.** (**A**) Representative overlaid DIC and GFP fluorescence images of day 1 adult wild-type, *drl-1(rhd109)*, and *drl-1(rhd109); fshr-1* double mutant animals (scale bar, 200 μm). (**B**) Body size of day 1 adult wild-type, *flr-2(ut5)*, and *fshr-1(ok778)* animals showing that both mutants are modestly smaller than wild-type (*, $P < 0.0001$, ns, not significant, one-way ANOVA). Raw data underlying panel B can be found in S16 Data.
(TIF)

**S9 Fig. Loss of intestinal DRL-1/FLR-4 hyperactivates the p38/PMK-1 signaling pathway.** (**A**) Western blot analysis of phospho-PMK-1, total-PMK-1, and actin levels in wild-type, presumptive kinase dead (KD) mutants, and null (0) mutants reared at 20 or 25°C. (**B**) Western blot analysis of phospho-PMK-1, total-PMK-1, and actin levels in *AID*::*drl-1* or *AID*::*flr-4* animals grown with or without 4 mM auxin (Intest., intestinal depletion, P*ges-1*::*TIR1*; Hyp., hypodermal depletion, P*col-10*::*TIR1*; Neur., pan-neuronal depletion; P*rgef-1*::*TIR1*). The *AID*::*drl-1* and *AID*::*flr-4* strains contain the *rhdSi42* transgene. Western blot experiments were performed twice with similar results. Raw images for panels A and B can be found in S4 and S5 Raw Images, respectively.
(TIF)

**S10 Fig. Reduced p38/PMK-1 signaling suppresses the effects of losing DRL-1/FLR-4.** (**A**) Representative overlaid DIC and mCherry fluorescence images (scale bar, 200 μm) and (**B**) body size (median and interquartile range; *, $P < 0.02$, **, $P < 0.001$, one-way ANOVA) of day 1 adult *drl-1(rhd109)* animals after knockdown p38/PMK-1 pathway components by RNAi. (**C**) Body size of P*ges-1*::*TIR1*; *mNG*::*3xFLAG*::*AID*::*flr-4* animals after simultaneous depletion of intestinal AID::FLR-4 (with 4 mM auxin) and knockdown of *pmk-1* or *fshr-1* by RNAi. Data are shown as the median and interquartile range (*, $P < 0.0001$, one-way ANOVA). Raw data underlying panels B and C can be found in S17 Data.
(TIF)

**S11 Fig. PKA hyperactivation does not stimulate p38 signaling.** (**A**) Body size of wild-type animals and the indicated mutants after control or *pmk-1* RNAi (day 1 adults; median and interquartile range; *, $P = 0.01$, **, $P < 0.0001$, unpaired *t* test). These mutations have been previously shown to activate PKA signaling (lf, loss-of-function; gf, gain-of-function). Quantification of the (**B**) total number and (**C**) absolute size of TIR-1 puncta in day 1 adult wild-type and mutant animals (mean +/− SD; n.s., not significant, one-way ANOVA). (**D**) A phospho-PMK-1 western blot analysis of whole animal lysates from wild-type animals and the indicated mutants. Actin was used as a loading control and the experiment was performed twice with

similar results. Raw data underlying panels A, B, and C can be found in S18 Data, and raw images for panel D can be found in S6 Raw Images.
(TIF)

**S12 Fig. DRL-1 inhibits the accumulation of nuclear PHA-4 through p38/PMK-1 signaling irrespective of diet.** (**A**) Fluorescence images (white arrowheads indicate intestinal nuclei; scale bar, 100 μm) and (**B**) quantification (median and interquartile range; n.s., not significant, **, $P < 0.0001$, one-way ANOVA) of PHA-4::GFP nuclear localization after depletion of intestinal AID::DRL-1 using 4 mM auxin in wild-type or *pmk-1(km25)* animals grown on *E. coli* HT1115. (**C**) A screenshot of PHA-4 ChIP-Seq data generated by the modENCODE project showing moderate binding of PHA-4 to the promoters of the *vit-3* and *vit-4* genes (black arrows) and strong binding of PHA-4 to the downstream gene *ckc-1* (red arrow). Raw data underlying panel B can be found in S19 Data.
(TIF)

**S1 Table. Identification of the *flr-2(rhd117)* mutation.** To identify the causative *drl-1 (rhd109)* suppressor mutations, EMS mutants were backcrossed to DLS364, the independently segregating F2 animals displaying the suppression phenotypes were pooled, the genomic DNA was sequenced, and candidate mutations were identified as described in the Materials and methods. The *flr-2* mutation (shown in bold) was selected for further analysis since it is predicted to be a strong loss-of-function allele. The resulting amino acid change is listed in the last column.
(TIF)

**S2 Table. Identification of the *fshr-1(rhd118)* mutation.** The candidate causative *drl-1 (rhd109)* suppressor mutations were identified as described in S1 Table and the Materials and methods. The *fshr-1* mutation (shown in bold) was selected for further analysis since it is predicted to be a strong loss-of-function allele. The resulting amino acid change is listed in the last column.
(TIF)

**S1 File. The *C. elegans* strains, crRNAs, and RT-qPCR primers used in this study.**
(PDF)

**S1 Data. Excel spreadsheet containing, in separate tabs, the numerical data underlying Fig 1C, 1D and 1E.**
(XLSX)

**S2 Data. Excel spreadsheet containing, in separate tabs, the numerical data underlying Fig 2B, 2C, 2D and 2E.**
(XLSX)

**S3 Data. Excel spreadsheet containing, in separate tabs, the numerical data underlying Fig 3B and 3C.**
(XLSX)

**S4 Data. Excel spreadsheet containing, in separate tabs, the numerical data underlying Fig 4B, 4C, 4D, 4E, 4F, 4G and 4H.**
(XLSX)

**S5 Data. Excel spreadsheet containing, in separate tabs, the numerical data underlying Fig 5B, 5C and 5E.**
(XLSX)

**S6 Data. Excel spreadsheet containing, in separate tabs, the numerical data underlying Fig 6C, 6D, 6E and 6F.**
(XLSX)

**S7 Data. Excel spreadsheet containing, in separate tabs, the numerical data underlying Fig 7B, 7C, 7D, 7E, 7G, 7H and 7I.**
(XLSX)

**S8 Data. Excel spreadsheet containing, in separate tabs, the numerical data underlying Fig 8B and 8D.**
(XLSX)

**S9 Data. Excel spreadsheet containing, in separate tabs, the numerical data underlying S1D and S1E Fig.**
(XLSX)

**S10 Data. Excel spreadsheet containing, in separate tabs, the numerical data underlying S2B and S2C Fig.**
(XLSX)

**S11 Data. Excel spreadsheet containing, in separate tabs, the numerical data underlying S3B, S3C and S3D Fig.**
(XLSX)

**S12 Data. Excel spreadsheet containing, in separate tabs, the numerical data underlying S4D, S4E and S4F Fig.**
(XLSX)

**S13 Data. Excel spreadsheet containing, in separate tabs, the numerical data underlying S5A, S5B and S5C Fig.**
(XLSX)

**S14 Data. Excel spreadsheet containing, in separate tabs, the numerical data underlying S6C and S6D Fig.**
(XLSX)

**S15 Data. Excel spreadsheet containing the numerical data underlying S7C Fig.**
(XLSX)

**S16 Data. Excel spreadsheet containing the numerical data underlying S8B Fig.**
(XLSX)

**S17 Data. Excel spreadsheet containing, in separate tabs, the numerical data underlying S10B and S10C Fig.**
(XLSX)

**S18 Data. Excel spreadsheet containing, in separate tabs, the numerical data underlying S11A, S11B and S11C Fig.**
(XLSX)

**S19 Data. Excel spreadsheet containing the numerical data underlying S12B Fig.**
(XLSX)

**S1 Raw Images. The raw images for Fig 3E.**
(PDF)

**S2 Raw Images. The raw images for S7A Fig.**
(PDF)

**S3 Raw Images. The raw images for S7B Fig.**
(PDF)

**S4 Raw Images. The raw images for S9A Fig.**
(PDF)

**S5 Raw Images. The raw images for S9B Fig.**
(PDF)

**S6 Raw Images. The raw images for S11D Fig.**
(PDF)

## Acknowledgments

The Caenorhabditis Genetics Center is supported by the NIH Office of Research Infrastructure Programs (P40 OD010440) and provided some of the strains used in this study. The total-PMK-1 antibody was generously provided by Dr. Read Pukkila-Worley (UMass).

## Author Contributions

**Conceptualization:** Sarah K. Torzone, Aaron Y. Park, Robert H. Dowen.

**Data curation:** Sarah K. Torzone, Aaron Y. Park, Peter C. Breen, Natalie R. Cohen, Robert H. Dowen.

**Formal analysis:** Sarah K. Torzone, Aaron Y. Park, Peter C. Breen, Natalie R. Cohen, Robert H. Dowen.

**Funding acquisition:** Robert H. Dowen.

**Investigation:** Robert H. Dowen.

**Methodology:** Sarah K. Torzone.

**Project administration:** Robert H. Dowen.

**Software:** Robert H. Dowen.

**Supervision:** Robert H. Dowen.

**Validation:** Robert H. Dowen.

**Visualization:** Robert H. Dowen.

**Writing – original draft:** Sarah K. Torzone, Robert H. Dowen.

**Writing – review & editing:** Sarah K. Torzone, Robert H. Dowen.

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
