## [Editor Report · Decision Letter 0]

5 Feb 2023

Dear Dr Dowen, 

Thank you for submitting your manuscript entitled "Follicle stimulating hormone signaling opposes the DRL-1/FLR-4 MAP kinases to balance p38-mediated growth and lipid homeostasis in C. elegans" for consideration as a Research Article by PLOS Biology.

Your manuscript has now been evaluated by the PLOS Biology editorial staff as well as by an academic editor with relevant expertise and I am writing to let you know that we would like to send your submission out for external peer review.

Once your full submission is complete, your paper will undergo a series of checks in preparation for peer review. After your manuscript has passed the checks it will be sent out for review. To provide the metadata for your submission, please Login to Editorial Manager (https://www.editorialmanager.com/pbiology) within two working days, i.e. by Feb 07 2023 11:59PM.

Kind regards,

Ines

--

Ines Alvarez-Garcia, PhD

Senior Editor

PLOS Biology

---

## [Decision Letter · Decision Letter 1]

21 Mar 2023

Dear Dr Dowen,

Thank you for your patience while your manuscript entitled "Follicle stimulating hormone signaling opposes the DRL-1/FLR-4 MAP kinases to balance p38-mediated growth and lipid homeostasis in C. elegans" was peer-reviewed at PLOS Biology. Please also accept my apologies for the delay in providing you with our decision. The manuscript has now been evaluated by the PLOS Biology editors, an Academic Editor with relevant expertise, and by three independent reviewers. 

As you will see, the reviews find the results interesting and think it is worth pursuing the manuscript for publication, however they also raise several concerns that would need to be addressed to confirm the conclusions. These include several controls, considering an alternative model that would fit the results better and several clarifications of the data.

In light of the reviews and discussions with the Academic Editor and the rest of the team, we would like to invite you to revise the work to thoroughly address the reviewers' reports. Among the points made by the reviewers, it will be particularly important to address Rev. 2's point 3 to clarify whether DRL-1 is itself a kinase or not with additional experiments. Regarding Rev. 3's comments, while it would be important to address points 12 and 13 as these refer to a potential discrepancy with a previously published paper and should be resolved, we would not require you to address points 6, 10 and 11.

Given the extent of revision needed, we cannot make a decision about publication until we have seen the revised manuscript and your response to the reviewers' comments. Your revised manuscript is likely to be sent for further evaluation by all or a subset of the reviewers.

**IMPORTANT - SUBMITTING YOUR REVISION**

3. Resubmission Checklist

a) *PLOS Data Policy*

b) *Published Peer Review*

d) *Blurb*

Please also provide a blurb which (if accepted) will be included in our weekly and monthly Electronic Table of Contents, sent out to readers of PLOS Biology, and may be used to promote your article in social media. The blurb should be about 30-40 words long and is subject to editorial changes. It should, without exaggeration, entice people to read your manuscript. It should not be redundant with the title and should not contain acronyms or abbreviations. For examples, view our author guidelines: https://journals.plos.org/plosbiology/s/revising-your-manuscript#loc-blurb

Sincerely,

Ines

--

Ines Alvarez-Garcia, PhD

Senior Editor

PLOS Biology

Reviewers' comments

Rev. 1:

The manuscript by Torzone, Park, and colleagues identifies the molecular and spatial components of signaling pathways that regulate vitellogenesis in C. elegans. Vitellogenins are lipoproteins through which hermaphrodite C. elegans transport lipids to their oocytes and embryos. This manuscript builds on previous findings that DRL-1, a kinase orthologous to the mammalian MEKK3, is a regulator of vitollegenesis. The new components identified and analyzed in this manuscript are i) FLR-4, a closely related MAPK that the authors show likely physically interacts with DRL-1, ii) FLR-2, a neuropeptide, iii) FSHR-1, a GPCR with similarity to glycoprotein hormone receptors such as the Follicle Stimulating hormone, and its predicted downstream signaling components (the canonical cAMP/PKA pathway functioning downstream of a G�s coupled receptor), and iv) PHA-4, a FOXA family transcription factor. The model proposed by the authors is that DRL-1 and FLR-4 kinases function in the C. elegans intestine, the tissue in which vitollegenins are formed, to promote expression of vit-3, encoding a subset of vitellogenin proteins and used as a marker of vitellogenesis here, along with growth rate and body size. In turn, neurally secreted FLR-2 acts on FSHR-1 to activate the intestinal PKA to counter this. They propose DRL-1/FLR-4 complex as well as PKA exert their effects by modulating TIR-1/SARM1, an upstream regulatory component of the p38 MAPK pathway. Finally, the p38 pathway is proposed to exert its effects, in part, through the PHA-4 transcription factor.

The area of investigation is interesting as there is relatively little known about the mechanisms that coordinate vitellogenesis, growth, and development. The model described above is mostly supported by genetic epistasis data as well as tissue specific rescue/knock-down experiments with several independent readouts. Additionally, there is some biochemical data to support the idea that FLR-4 and DRL-1 physically interact. Although many of the regulatory interactions remain poorly understood at a biochemical or molecular level, the conclusions are well supported by the presented data and the findings constitute a real advance for identifying new components that coordinate vitellogenesis.

Although some of the regulatory interactions of this manuscript were either already known or easily predictable (such as the role of FLR-4 and the p38 pathway), other components are novel and emerged from unbiased screens (such as the role of the FL-2/FSHR-1 pathway). Even in the context of the predicted relationships, the authors have made advances by developing reagents that allowed for more rigorous analyses (such as tagging of endogenous loci; tissue specific depletions of signaling components). Compared to the rest of the manuscript, the data pertaining to the role of PHA-4 are less convincing, but the authors appropriately describe the role of PHA-4 as being partial (inclusion of the PHA-4 data in the manuscript is justified).

Overall, the manuscript is quite solid, well-written, and, as indicated, an advance. There are a few, minor concerns with the manuscript, most of which can be remedied by changes/additions to the writing:

1) As the authors nicely describe, vitellogenesis is upregulated at a very specific stage of development. What is not very clear from the data is the temporal aspects, if any, of the described signaling pathways. The manuscript would be strengthened if such data could be provided, for example, is secretion of FLR-2 developmentally regulated? This data is not necessary for publication but would strengthen the manuscript. In the absence of data, a clearer discussion of this point should be included.

2) Similar to point 1, data OR a clearer discussion on the temporal roles of DLR-1/FL-4 would be helpful: given that their losses slow down development, is the loss of vit-3 expression a secondary consequence of this slowed down development? Same applies to PHA-4.

3) While the introduction section of the manuscript nicely described vitellogenesis and the vit-3 reporter is a key reagent in the manuscript, most of the rest of the manuscript, especially the discussion section, seems to move away from vitellogenesis and instead use the more generic "lipid homeostasis" as a way of describing the data. Of course, vitellogenesis affects "lipid homeostasis" but the authors appear to almost treat lipid homeostasis and vitellogenesis as two unrelated processes when discussing their results. The intestinal lipids are the obvious source of lipids for the vitellogenin lipoproteins. The manuscript would be improved by being clearer about this.

Rev. 2:

The manuscript by Torzone et al reported signaling pathways that potentially worked in parallel (Follicle stimulating hormone-like) or downstream (p38 MAP kinase) of DRL-1, a protein that shares sequence similarity with mammalian mitogen-activated protein kinases. Based on results from forward genetic screens and targeted RNAi screens in the drl-1 mutant background, the authors proposed the convergence of cell-autonomous and non-cell-autonomous signals at the C. elegans intestine, which ultimately regulates the expression of a key lipoprotein gene vit-3. The same signaling network also appeared to regulate larval development and adult lifespan. Using CRISPR-based editing, this study created a large set of new transgenic lines that bore precise mutations and/or fluorescent protein/epitope tags. These lines allowed the authors to study proteins of interest without over-expression, which has the potential of clarifying a large body of work in a crowded field. However, to achieve this goal, the authors should demonstrate that the fluorescent protein/epitope tagged versions of DRL-1 and FLR-4 proteins retain full function. This is especially important because certain fluorescent proteins or epitope tags are known to perturb expression or function in specific contexts (https://doi.org/10.1371/journal.pone.0183067).

Major points:

1. [Line 3] "Follicle stimulating hormone signaling" in the title is slightly misleading because FSH does not exist in C. elegans. The authors may consider using "FSH-like signaling" instead.

2. [Line 133] The use of col-10 promoter to drive transgene expression in the hypodermis may be flawed in this study. This is because col-10 is strongly expressed in larvae but not adults. Yet data reported in the manuscript (RNAi, and auxin-mediated protein degradation) were collected from day 1 adult animals. Although this may not affect the conclusion that DRL-1 acts in the intestine, the authors should consider alternative controls.

3. [Line 154] The definition of DRL-1 as a stand-alone MAP kinase is problematic. As stated by the authors, key residues for ATP binding and catalysis are missing in DRL-1. Although a prior publication (https://doi.org/10.1111/acel.12218) from another lab reported the ability of DRL-1 to phosphorylate a model substrate (Maltose Binding Protein), the results could be interpreted differently. Based on the method described, it was possible that proteins (including another kinase) that co-immunoprecipitated with DRL-1 from mammalian cells could be responsible for substrate phosphorylation. Therefore, the authors should consider an alternative model for DRL-1 action. Is it possible that DRL-1 is a non-catalytic partner of FLR-4? Is the temperature sensitive phenotype of the DRL-1(P269S) 'activation loop' mutant caused by structural instead of catalytic defect? Ideally, kinase assays would need to be performed with purified DRL-1 proteins, if the authors wish to draw a strong conclusion on its role as a kinase.

4. [Line 179] The diet-dependent lifespan extension of DRL-1 or FLR-4 deficient animals was difficult to follow. The authors should provide additional background information and spell out the significance of their results.

5. [Line 205] The use of genetic mutant or RNAi to perturb the formation of lysosome related organelles (LROs) and reduce intestinal autofluorescence signals is an interesting approach, with caveats. The authors assumed that the block in LRO formation did not contribute to the localization of DLR-1 and FLR-1. Alternative imaging techniques for detecting weak fluorescence signals and subtracting autofluorescence should be attempted. In addition, co-localization of DRL-1 and FLR-4 cannot be used to address the statement in [Line 211]: "to test whether DRL-1 and FLR-4 form a protein complex".

6. [Line 215] To properly assess the efficiency of immunoprecipitation (enrichment factor), the authors should state the amount of whole cell lysate loaded in relation to the amount of lysate used for immunoprecipitation, i.e. % input. Based on Fig. 3E, lower panel, FLR-4 was less efficient than DRL-1 in pulling down DRL-1. The authors should elaborate on the significance of this result.

7. [Line 390] Similar to point 4, the use of glo-3 RNAi to reduce auto-fluorescence for TIR-1::wrmScarlet imaging is not free of caveats. Separately, although it is not the authors' burden to demonstrate that TIR-1::wrmScarlet puncta formation equates phase transition, please consider toning down [Line 392] "loss of drl-1 induces TIR-1::wrmScarlet phase transition". There is a similar problem with [Lines 36-37]. Have TIR-1 puncta been detected when TIR-1 was fused to different fluorescent protein partners?

8. [Line 402] Since pha-4 RNAi could regulate Pvit-3::mCherry reporter activity in drl-1 mutant, are PHA-4 binding sites found in the vit-3 promoter? The images in Fig. 8C and Fig. S9A were hard to follow. Although the method for measuring PHA-4::GFP nuclear signal was described in the Materials and methods section, readers who are unfamiliar with C. elegans anatomy would be confused by the images at such low magnification. The addition of magnified insets with nuclei marked should be considered.

Rev. 3:

Torzone et al. uses C. elegans to identify the two opposing, inhibitory and stimulatory, effects on the p38 MAPK pathway that regulates metabolic homeostasis during development. They show that mutations in the kinases, DRL-1 and FLR-4 that seem to form a complex on the intestinal membrane, engage the p38 MAPK to prevent vitellogenesis and reduce growth. In a forward genetic screen, they identify the FLR-2 (neuronal)/FRSH-1 (intestinal) axis as the neurohormonal signaling pathway that stimulates the p38 MAPK through a G-protein signaling, leading to phase transition of p38 MAPK upstream modulator, TIR-1. Together, the authors uncover an intricate signaling network that regulates organismal growth and development. However, it should be noted that both FLR-4 and DRL-1 do not have direct orthologs in other systems and may be restricted to a nematode sub-clade. So, the global appeal will reduce if the context of the study is not well defined. It is not clear under what circumstances the opposing pathways work; what are the individual triggers for the two arms.

While a part of the work is quite interesting, the study seems premature in multiple aspects:

Major comments:

1) This reviewer is not convinced by the model that the study has proposed. Is it not possible that flr-2 works downstream of drl-1/flr-4 that negatively regulates it? The alternative possibilities should be tested and eliminated.

2) What are the conditions that will activate flr-2/fshr-1? The authors should show what happens to TIR-1 phase transition when they activate that arm independently.

3) The authors fail to establish the flr-2/fshr-1 axis. While they are independently shown to modulate p38 MAPK, there is no data suggesting that they are in the same pathway. Same is the case with the G protein signaling.

4) When showing neuron-specific flr-2 depletion, they need to show intestine-specific depletion as a control. Vice versa for fshr-1. Same for tissue-specific RNAi for the G-protein signaling.

5) The authors should confirm that the kinase dead version of the proteins that they have generated are indeed biochemically deficient in kinase activity.

6) The authors should show kinase activity when they refer to the hyperactivation of the p38 MAPK.

7) The authors claim that the two proteins form a complex on the membrane using colocalization and co-IP. While the co-IP data is convincing, the colocalization is not. The nature of the complex is also not studied. Will overexpression/activation of one protein suppress the other?

8) Line 237: does the drl-1;flr-2 suppress the increased life span of drl-1?

9) Line 348: The authors get only partial suppression of phenotype. Have they checked sek-1 or nsy-1 mutants? Is it possible that other PMKs are required?

10) The phase transition of TIR-1 is not unexpected. They should have used the oligomerization defective or enzymatic action defective versions as control.

11) The PHA-4 data is incomplete without proper understanding of what genes the transcription factor regulates in this context. How is the transcription factor activated?

12) The authors have observed severe developmental defects in flr-4(n2259) worms. This contradicts the study from Verma et al, 2018 and Nair et al, 2021. Authors should comment on this.

13) In line 180, the statement that flr-4 RNAi increases life span only on HT115, but not OP50 is incorrect. In that paper (Verma et al 2018), the authors show that life span is extended on both HT115 and OP50 flr-4 RNAi; this was not the case with the n2259 allele.

---

## [Decision Letter · Decision Letter 2]

2 Aug 2023

Dear Dr Dowen,

My name is Luke Smith - I am an editor at PLOS Biology, and I am contacting you on behalf of my colleague, Ines, who is away on vacation this week. I am handling your manuscript "Glycoprotein hormone signaling opposes the DRL-1/FLR-4 MAP kinases to balance p38-mediated growth and lipid homeostasis in C. elegans" while Ines is away. Thank you for your patience while your revised manuscript was evaluated as Research Article at PLOS Biology. Your revised study has now been evaluated by the PLOS Biology editors, the Academic Editor and the original reviewers. 

As you will see in the reviews, at the end of this email, the reviewers are largely satisfied by the changes made in this revision. However, the reviewers have a number comments and lingering concerns that we think will need to be addressed before publication. 

After discussion with the Academic Editor, we think that all the remaining reviewer points, including those of Reviewer #3 should be addressable by changes to the text and should not require further experiments. We agree with Reviewer #3 that the western blot replicates and other source data they request should be provided by the authors. Regarding the specific point of Reviewer #3 about controls for the phase transitions (originally R3, point 10), the Academic Editor has commented that while these would strengthen the study, on balance, these are not necessary for publication. However, we think you will need to address this point directly and explicitly by discussing in the main manuscript text how your results and those of the Pukkila-Worley group (Elife) differ and how they might be reconciled.

In total, we are pleased to offer you the opportunity to address the remaining points from the reviewers in a revision that we anticipate should not take you very long. We will then assess your revised manuscript and your response to the reviewers' comments with our Academic Editor aiming to avoid further rounds of peer-review, although might need to consult with the reviewers, depending on the nature of the revisions.

**IMPORTANT: As you address these last comments, please also attend to the following editorial requests: 

1) TITLE: After some discussion within the team, we think the title could be streamlined a bit. Therefore, if you agree, we suggest you change the title to "Opposing action of glycoprotein hormone and DRL-1/FLR-4 MAP kinases balance p38-mediated growth and lipid homeostasis in C. elegans"

2) BLURB: In the relevant portion of our online system, please provide a blurb which (if accepted) will be included in our weekly and monthly Electronic Table of Contents, sent out to readers of PLOS Biology, and may be used to promote your article in social media. The blurb should be about 30-40 words long and is subject to editorial changes. It should, without exaggeration, entice people to read your manuscript. It should not be redundant with the title and should not contain acronyms or abbreviations.

3) BLOTS AND GELS: Please note that we require the original, uncropped and minimally adjusted images supporting all blot and gel results reported in an article's figures or Supporting Information files. We will require these files before a manuscript can be accepted so please prepare and upload them now. Please carefully read our guidelines for how to prepare and upload this data: https://journals.plos.org/plosbiology/s/figures#loc-blot-and-gel-reporting-requirements

>>Please provide the uncropped and minimally adjusted images relating to Fig 3E; Fig S7A-B; Fig S9A-B; Fig S11D

4) DATA: You may be aware of the PLOS Data Policy, which requires that all data be made available without restriction: http://journals.plos.org/plosbiology/s/data-availability. For more information, please also see this editorial: http://dx.doi.org/10.1371/journal.pbio.1001797

a. Supplementary files (e.g., excel). Please ensure that all data files are uploaded as 'Supporting Information' and are invariably referred to (in the manuscript, figure legends, and the Description field when uploading your files) using the following format verbatim: S1 Data, S2 Data, etc. Multiple panels of a single or even several figures can be included as multiple sheets in one excel file that is saved using exactly the following convention: S1_Data.xlsx (using an underscore).

b. Deposition in a publicly available repository. Please also provide the accession code or a reviewer link so that we may view your data before publication. 

>>Regardless of the method selected, please ensure that you provide the individual numerical values that underlie the summary data displayed in the following figure panels as they are essential for readers to assess your analysis and to reproduce it:

Fig 1C-E; Fig2B-E; Fig 3B-C; Fig 4B-H; Fig 5B-C,E; Fig 6C-F; Fig 7B-E,G-I; Fig 8B,D

Fig S1D-E; Fig S2B-C; Fig S3B-D; Fig S4D-F; Fig S5A-C; Fig S6C-D; Fig S7C; Fig S8B; Fig S10B-C; Fig S11A-C; Fig 12B

>>>Please also ensure that figure legends in your manuscript include information on where the underlying data can be found, and ensure your supplemental data file/s has a legend.

>>Please ensure that your Data Statement in the submission system accurately describes where your data can be found.

**IMPORTANT - SUBMITTING YOUR REVISION**

*Resubmission Checklist*

*Published Peer Review*

Sincerely,

Luke

Lucas Smith, PhD

Senior Editor

PLOS Biology

lsmith@plos.org

--on behalf of--

Ines Alvarez-Garcia, PhD

Senior Editor

PLOS Biology

REVIEWS:

Reviewer #1: The authors have adequately addressed all of my concerns. Congratulations on a nice piece of science.

Reviewer #2: The authors have satisfactorily addressed my concerns. 

Minor comments:

1. Please add arrowheads to specify nuclear GFP signals in Fig. 8C and S12A. 

2. [Line 573] All *stains* used in this study are listed in S1 File. Please correct typo. 

Reviewer #3: The authors address most of my comments satisfactorily and the manuscript has improved quite a lot. Following are my observations:

The phase transition images are still not convincing to this reviewer and authors should consider placing high-resolution images as a supplementary figure. In the publication from Pukkila-Worley group in Elife that reported the phase transition, the number of puncta increases whereas here it does not, and only the size increases. How this may be reconciled? So, the use of the controls that were suggested in the first round of review (reviewer 3, point 10) is important to perform. 

In the Figure 6B and C, kin-1 RNAi does not reactivate Pvit-3::mcherry in the drl-1 mutant. However, the kin-1 can increase the body size. Since it is the terminal molecule of the pathway that authors propose to communicate to TIR-1, how is this point reconciled? 

In figure S8A, line 344, the representative image of nsy-1, sek-1 doesn't show reactivation of Pvit-3::mcherry in the drl-1 mutant while the quantification shows that. Why is it so?

It is not clearly mentioned how many times the western blot experiments of Fig S7 were performed. Same with the IP experiment in Fig 3E. It will be great if the authors present all the replicates in the source data file for the readers to understand the variability they may expect.

In the figure 3E, is a band expected in the WCL of FLAG western? If yes, why is it not visible here? 

I could not find the source data file for the experiments. If it is not attached to the revised document, the authors may consider putting all primary data in a source file.

In the legend, it will be great if the authors clearly mention the number of replicates (technical and biological) used for each experiment.

---

## [Editor Report · Decision Letter 3]

2 Sep 2023

Dear Dr Dowen,

Thank you for the submission of your revised Research Article entitled "Opposing action of the FLR-2 glycoprotein hormone and DRL-1/FLR-4 MAP kinases balance p38-mediated growth and lipid homeostasis in C. elegans" for publication in PLOS Biology. On behalf of my colleagues and the Academic Editor, Alex Gould, I am delighted to let you know that we can in principle accept your manuscript for publication, provided you address any remaining formatting and reporting issues. These will be detailed in an email you should receive within 2-3 business days from our colleagues in the journal operations team; no action is required from you until then. Please note that we will not be able to formally accept your manuscript and schedule it for publication until you have completed any requested changes.

PRESS

Sincerely, 

Ines

--

Ines Alvarez-Garcia, PhD

Senior Editor

PLOS Biology
